# Low Pt loading for high-performance fuel cell electrodes enabled by hydrogen-bonding microporous polymer binders

Hongying Tang[1,2,3] ✉, Kang Geng[1], David Aili[3], Qing Ju[1], Ji Pan[4], Ge Chao[1], Xi Yin[1], Xiang Guo[1], Qingfeng Li[3] ✉ & Nanwen Li[1] ✉

A key challenge for fuel cells based on phosphoric acid doped poly-benzimidazole membranes is the high Pt loading, which is required due to the low electrode performance owing to the poor mass transport and severe Pt poisoning via acid absorption on the Pt surface. Herein, these issues are well addressed by design and synthesis of effective catalyst binders based on polymers of intrinsic microporosity (PIMs) with strong hydrogen-bonding functionalities which improve phosphoric acid binding energy, and thus preferably uphold phosphoric acid in the vicinity of Pt catalyst particles to mitigate the adsorption of phosphoric acid on the Pt surface. With combination of the highly mass transport microporosity, strong hydrogen-bonds and high phosphoric acid binding energy, the tetrazole functionalized PIM binder enables an $H_2$-$O_2$ cell to reach a high Pt-mass specific peak power density of 3.8 W $mg_{Pt}^{-1}$ at 160 °C with a low Pt loading of only 0.15 $mg_{Pt}$ $cm^{-2}$.

High-temperature proton exchange membrane fuel cell (HT-PEMFC) systems based on phosphoric acid-doped polybenzimidazole (PBI/PA) membrane operate between 140 to 180 °C. Applications of the technology covers distributed power generation as well as automotive range extension[1]. Technical features of sSuch systems have advantages include simplified water and heat management, high tolerance to fuel impurities e.g. CO and atmospheric contamination[2], as well as better utilization of the waste heat[3,4]. A major challenge is the relatively low power output of the technology due to the sluggish kinetics of the oxygen reduction reaction and low utilization of noble-metal catalysts. In the presence of phosphoric acid electrolyte, the Pt surface involves the strong adsorption of acid molecules ($H_3PO_4$) in the low potential range (300–400 mV) and acid anions ($H_2PO_4^-$) at intermediate potential range (700–800 mV) during the cell operation at 160 °C[5–8]. As a result, the state-of-the-art technology uses electrodes with a high platinum loading in the range 0.5–1.0 $mg_{Pt}$ $cm^{-2}$, typically around 0.7 $mg_{Pt}$ $cm^{-2}$, as recently reviewed[1], which in combination with the relative low

power output leads to the increased materials and construction cost of the technology.

Improvement of the catalyst performance needs optimization of the catalyst layer where the electrochemical reactions occur. For noble metal catalyst sites to be electrochemically active the catalyst layer must be conductive for both electrons and protons, which are produced or consumed by the electrochemical reaction at the sites. At the same time, the layer needs to be permeable to the gaseous reactants as well as to the product water. The essence of the catalytic layer is the three-phase boundaries where the loose Pt/C particles are bonded together by means of polymeric binders, as schematically represented in Supplementary Fig. 1. The catalysts consist of platinum metal particles in size of 2–5 nm supported on carbon black particles of ca. 50 nm. The Pt loading is normally in a high metal 40–60 wt% Pt range in order to reduce the over catalyst layer thickness and hence the mass transportation limitation. The carbon particles usually form aggregates with relatively strong bonding while the aggregates are further forming agglomerates by the weak coulombic attraction. As a result,

[1]State Key Laboratory of Coal Conversion, Institute of Coal Chemistry, Chinese Academy of Sciences, Taiyuan, China. [2]Tianjin Key Laboratory of Water Resources and Environment, Tianjin Normal University, Tianjin, China. [3]Department of Energy Conversion and Storage, Technical University of Denmark, Elektrovej, Building 375, 2800 Lyngby, Denmark. [4]College of Chemistry, Chemical Engineering and Materials Science, Soochow University, No. 199 Renai Road, Suzhou, China. ✉e-mail: hytang@tjnu.edu.cn; qfli@dtu.dk; linanwen@sxicc.ac.cn

the catalyst layer contains two types of pores, the primary (small) pores of <10 nm size between carbon particles within aggregates and secondary (large) pores of 10–100 nm size between the aggregates of the agglomerates[8].

For low temperature PEMFCs, the proton conductivity is provided by impregnation of the catalyst layer with perfluorosulfonic acid (PFSA) ionomer, which fills in the primary pores and becomes conductive when swollen with water during the fuel cell operation[9]. The Pt loading in the low temperature PEMFC is practically about 0.3 mg$_{Pt}$ cm$^{-2}$ and targeting at 0.15 mg$_{Pt}$ cm$^{-2}$, with an estimated Pt utilization as high as 70–85%[10–12]. In HT-PEMFCs, the commonly used binder includes PBI, polytetrafluoroethylene (PTFE) and poly-vinylidene fluoride (PVDF)[13–16], which facilitate the establishment of the triple phase boundaries by the acid diffusion from the membrane. Several issues are raised in connection to the use of these binders. First of all, they block a fraction of the catalyst sites leading to poor utilization of Pt. Secondly, all these binder materials are in general a dense phase, into which the reactant gases from the secondary pore channels are difficultly to dissolve, diffuse and reach the catalytic sites as shown in Supplementary Fig. 1. Compared with the PFSA ionomer at 80 °C, the product of the oxygen solubility and diffusion coefficient is at least one order of magnitude lower in concentrated phosphoric acid and acid doped PBI membranes at 150 °C[17]. It is hence detrimental to both the electrode kinetics and mass transport performance of the electrodes. Only around 15% Pt utilization ihas been reported in literature for electrodes of PBI/PA membranes based HT-PEMFCs apparently due to the lack of iono-mer/binder with excellent ability of PA retention to mitigate PA adsorption on Pt surface as well as good mass transport ability and proton conductivity in the MEAs[11,16]. Most recently, the intrinsically proton conductive phosphate/sulfonated polymers have been developed as ionomers for HT-PEMFC and better cell performance has been achieved for the MEAs with these conductive binder materials[18–20], however, higher catalyst loadings of >0.5 mg$_{Pt}$ cm$^{-2}$ are still necessary.

The strategy of the present work is to design and synthesize effective binder materials possessing (1) substantial microporosity to enhance the mass transport and (2) acid interactive functionalities via hydrogen-bonding and/or acid-base interaction to preferably retain PA and alleviate the flooding and adsorption on the Pt surface, as shown in Fig. 1. A set of criteria are proposed including gas permeability, phosphoric acid binding energy, hydrogen binding energy, acid wetting and uptake. The molecular design is based on polymers of intrinsic microporosity (PIMs) and a series of selected functional groups from which effective binders are synthesized and characterized. The candidate binder, tetrazole modified PIM, is used for manufacturing gas diffusion electrodes and evaluated in fuel cell tests showing super performance in terms of specific power density and platinum utilization.

## Results and discussion

### PA binding energy as a criterion for screening of the functionality groups

PIMs, as first reported by Budd and McKeown et al.[21,22]., are characterized by the inefficient packing of the macromolecules provided by rigid and contorted polymer chains. The resultant polymers contain a continuous network of interconnected intermolecular voids i.e. exhibiting high intrinsic microporosity allowing both gaseous and liquid phases to meet at the active sites within the electrodes which is essential as the effective catalyst binder[23–25]. Aiming at tuning the acid affinity, PIMs are further modified by introduction of a variety of functional groups (Supplementary Fig. 2), which are screened by the intermolecular interaction energy, PA binding energy ($E_{PA-binding}$), between the repeat unit in the binder materials and PA as the first criterion. The PA adsorption energies on the Pt catalyst surface ($E_{adsorption}$), have been calculated using CASTEP (Cambridge serial total energy package) program module to be 19.5 kcal/mol (Supplementary Fig. 3 and Data 1)[26]. The PA binding energy of the repeated units in the polymers was calculated and found to be 3.4 kcal mol$^{-1}$ for PTFE, 10.3 kcal mol$^{-1}$ for PBI (as listed in Table 1, Supplementary data 1). As a result, the PA in the catalyst layer, transferred from PA-doped membranes during fuel cell operation, preferentially adsorbs on the Pt catalyst, which hinders the kinetics of oxygen reduction and limits the mass transport of the oxidant (as shown in Fig. 1a, b)[1,27].

It is expected that the functionalized PIMs will enable the effective permeation of reactant gases due to the intrinsic microporosity and, at the same time, exhibit the strong interaction with the PA. The firm retention of the PA preferably around the binders will alleviate the flooding and adsorption of the PA on the Pt surface as shown in Fig. 1c. For screening the functional groups, their interaction between the

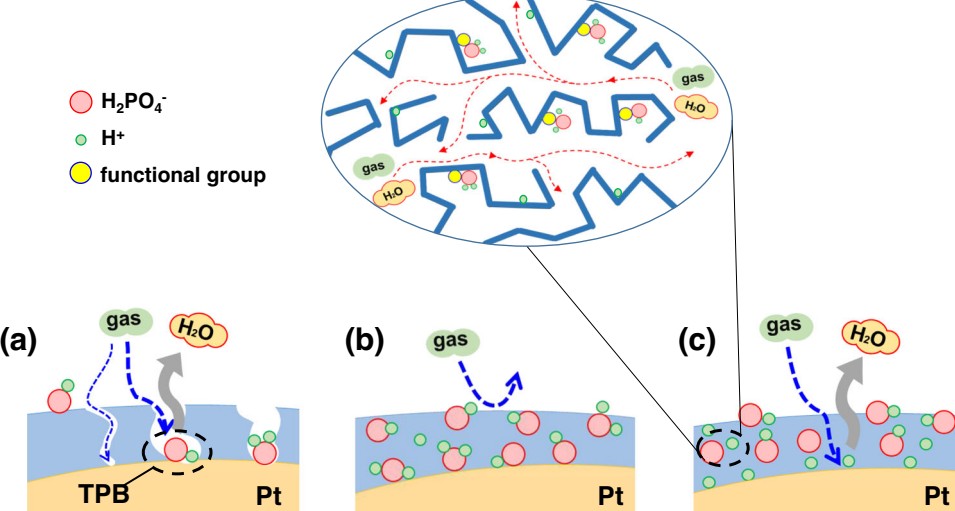

**Fig. 1 | Alleviation of Pt catalyst poisoning from PA flooding and enhancement of electrochemical reactivity and Pt utilization at ionic porous-structured interfaces. a** At a hydrophobic porous catalysts/electrode surface with binders having less proton conductivity and lower PA binding energy such as PTFE, **b** at an acidophilic catalysts/electrode surface using binders with low PA binding energy and less porous structure such as PBI, **c** hydrogen-bonding dependent functionalized PIMs (PIM-Tz, PIM-AO, e.g.) with high PA binding energy based porous-structured catalyst layer.

**Table 1 | The properties of PIM-1 and its variants. PTFE and *m*PBI are included for comparison**

| Polymers | $M_n$ (kDa) | $pK_a$ [a] | $E_{H\text{-}bonding}$ (kcal mol$^{-1}$) [a] | $E_{PA\text{-}binding}$ (kcal mol$^{-1}$) [a] | $S_{BET}$ [m$^2$ g$^{-1}$] [b] | Permeability (Barrer)[c] | |
|---|---|---|---|---|---|---|---|
| | | | | | | $H_2$ | $O_2$ |
| PIM-Tz (75%) | 41 | 4.3 | 3.9 | 25.4 | 431 | 1920 | 498 |
| PIM-COOH | 40 | 3.7 | 2.8 | 15.0 | 579 | 400 | 108 |
| PIM-AO | 53 | 13.3 | 3.4 | 23.3 | 531 | 920 | 151 |
| PIM-CONH$_2$ | 42 | 8.8 | 3.0 | 14.1 | 543 | 620 | 135 |
| PIM−1 | 70 | N.A. | 2.1 | 12.8 | 693 | 3350 | 960 |
| PTFE | N.A. | N.A. | 2.5 | 3.4 | N.A. | 10.5[63] | 5[63] |
| *m*PBI | 50 | 12.7 | 2.2 | 10.3 | 2 | 1.73[64] | 0.09[64] |

[a]$pK_a$ value of the related functional groups in PIMs, hydrogen-bonding energy ($E_{H\text{-}bonding}$) and PA binding energy ($E_{PA\text{-}binding}$) were obtained via calculation.
[b]N2 adsorption isotherm at 77 K of the PIMs in the powder form.
[c]H2 and O2 permeability of dense PIM films.

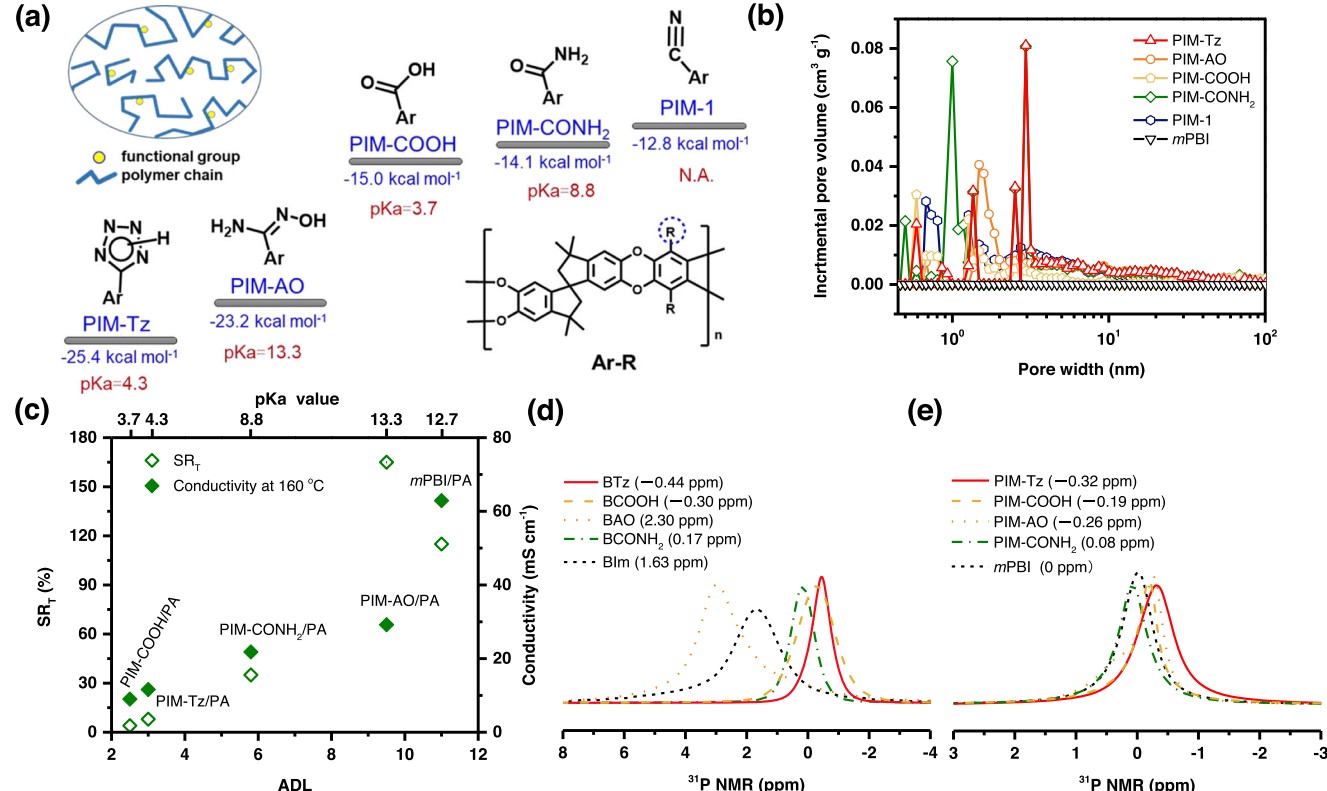

**Fig. 2 | Properties of the functional PIMs. a** Structures, $pK_a$ values and phosphoric acid binding energies of PIMs and its variants; **b** incremental pore volumes of PIMs and PBI as a function of the pore size measured from N$_2$ adsorption/desorption isotherms. **c** Characteristics of PA doped films of the functionalized PIMs as well as PBI including the acid doping level (ADL) and thickness swelling ratio (SR$_T$) at room temperature and proton conductivity at 160 °C. Solid-state $^{31}$P NMR spectra of PA-doped samples: **d** functional benzene monomers doped with 1 equivalent pure PA and (**e**) polymer powders saturated with pure PA. Chemical shifts are presented in parentheses, externally referenced to (NH$_4$)$_2$HPO$_4$ solid.

repeat units in functional PIMs and PA is evaluated in terms of the PA binding energy. The selected functional groups include amide, amidoxime, carboxylic and tetrazole, as shown in Fig. 2a and Supplementary Fig. 2, the PA binding energy of which was found to range from 15.0 to 25.4 kcal mol$^{-1}$, higher than that of the repeat units in PTFE and PBI. Most important, some functional PIMs, e.g. PIM-Tz and PIM-AO, among others, have higher PA binding energy than the adsorption energy of PA on Pt surface, hinting the potential of these functionalities to preferably uphold the acid in the vicinity of the Pt catalyst particles.

## Functionalization of PIMs

According to the PA binding energy as discussed above, PIM−1 was post-modified via the -CN with the selected functional groups (Supplementary Fig. 2)[21–23,28]. The obtained functionalized polymers are amide-PIM-1 (PIM-CONH$_2$, degree of functionalization (DF): 100%), amidoxime-PIM-1 (PIM-AO, DF: 100%), carboxylic-PIM (PIM-COOH, DF: 100%) and tetrazole-PIM-1 (PIM-Tz, DF: 75%). As shown in Fig. 2a and Supplementary Fig. 4, the degree of the functionalization for PIM-CONH$_2$, PIM-AO and PIM-COOH is determined by the intergral ratio of functional hydrogen to aromatic hydrogen in the polymers via the analysis of $^1$H NMR spectra. For PIM-Tz, the DF is assessed by the reaction conditions according to the literature due to the difficult analysis of $^1$H NMR spectra resulting from the strong intermolecular hydrogen bonds in the polymer matrix[23]. PIM-Tz with DF100% having a BET area of about 30 m$^2$/g was initially tested which showed poor performance due to the strong hydrogen bonding and the resultant

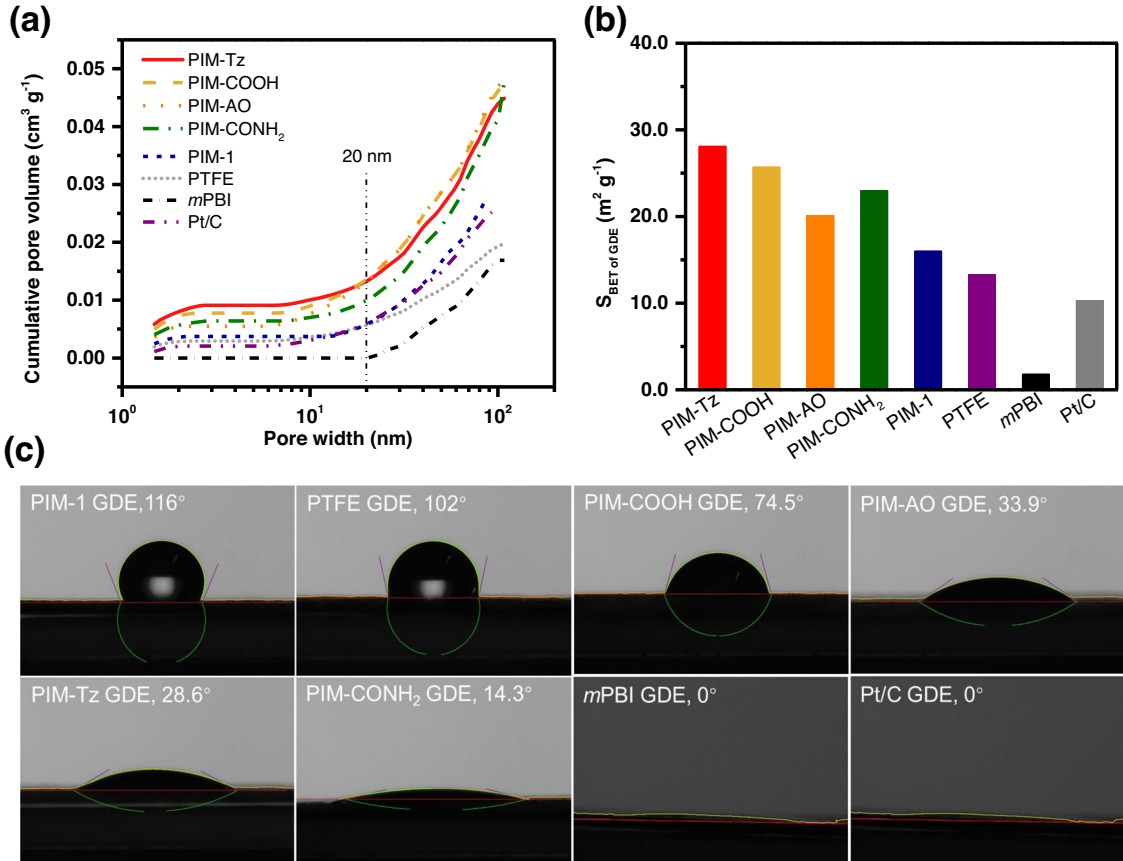

**Fig. 3 | Characteristics of the GDEs with PIMs as binders in the catalyst layer.** **a** Pore-size distributions of GDEs with various binders as indicated in the figure. **b** $S_{BET\ of\ GDE}$ of GDEs with different binders measured by $N_2$ adsorption isotherm at 77 K. **c** The contact angle of the different GDEs with phosporic acid. The Pt/C GDE means refers to binderless electrodes. The content of the binder in the catalyst layer was 20 wt% of total solid in the catalyst ink.

low porosity or gas permeability. In the following work a relatively low (75%) DF is used for PIM-Tz. During the modification process, PIM-1 degrades due to the severe reaction conditions e.g. strong acidity or high temperature as indicated by the slightly decreased molecular weight ranging from 40 to 53 kDa (Table 1). The thermogravimetric analysis (Supplementary Fig. 5) shows that the PIMs are thermally stable enough to withstand high temperatures for HT-PEMFC application. The functionalized polymers in form of powders were analyzed by the nitrogen gas adsorption isotherms and showed a decline of the Brunauer-Emmett-Teller (BET) surface area, as seen from Table 1, probably due to the hydrogen-bonding interaction leading to tighter chain packing of polymer chain. Compared with *m*PBI, these polymers have BET areas of 200–300 times higher, indicating the microporous nature of the materials. Of these, PIM-Tz shows the lowest BET value of 431 $m^2\ g^{-1}$, likely due to the strong interchain interactions through the tetrazole sites[29].

A significant proportion of micropores in these functionalized PIMs ranges from 0.2 to 3 nm, as revealed by the micropore analysis using Original Density Functional Model (Fig. 2b). As a base line, PBI has little porosity in the pore size range of up to 100 nm. The microporosity of these polymers permits the gas penetration easily, and is assessed by the measurement of the $H_2$ and $O_2$ permeabilities of self-supported PIM films using a constant-volume permeation testing approach. The model polymer (PIM-1) film exhibits the highest permeability for hydrogen and oxygen, in good agreement with literature[21,22]. All functionalized PIMs showed also significantly higher permeabilities for hydrogen and oxygen. As a comparison, PTFE and *m*PBI, exhibit $H_2$ and $O_2$ permeabilities of 2–3 orders of magnitude lower than that of e.g. the functionalized PIMs.

## Hydrogen-bonding interactions between functionalized PIMs and PA

Phosphoric acid has an infinitive network of hydrogen-bonds, which have the bond strength in an intermediate range from 2.3 to 7.2 kcal $mol^{-1}$, allowing for frequent breaking and forming of hydrogen-bonds, and henceforth the dominating Grotthuss mechanism of the proton conductivity for 100% PA. In case of PA doped PBI membranes, more than 96% of the Grotthussian conductivity was recently reported[30]. The hydrogen-bonding energy ($E_{H\text{-}bonding}$) and p$K_a$ of the functional groups (Brønsted acid/base) are essential for the interaction between these functionalities and PA. Thus, the p$K_a$ value of the functional groups and the hydrogen-bonding energy in the PIMs with 85% PA, in addition to the above discussed PA binding energy, were calculated (Supplementary Fig. 3 and Supplementary Data 1). It is noted that the PA binding energy of the functional repeated units in PIMs having stronger hydrogen-bonding capabilities (e.g. PIM-Tz and PIM-AO) are much higher than those having strong acid-base interaction with PA (e.g. higher p$K_a$ value). The p$K_a$ value of the functional groups was calculated using Relative Gibbs free energy change (RGC) approaches and found to vary in a wide range from 3.7 to 13.3. As shown in Fig. 2 and Table 1, the functional group in PIM-AO exhibits the highest p$K_a$ value, which is close to that of benzimidazole groups in *m*-PBI. As the p$K_a$ of PA is 2.12, this high p$K_a$ implies the strongest acid-base interactions of PIM-AO with PA, which was verified also by the higher acid doping level (ADL) and swelling ratio of the corresponding membranes as shown in Fig. 2c. On the other hand, PIM-Tz has a low p$K_a$ value but shows the highest hydrogen-bonding energy of 3.9 kcal $mol^{-1}$ based on Dreiding in the whole MD simulations (Supplementary Data 2). This value is much higher than that of *m*PBI as well as other functionalized PIMs

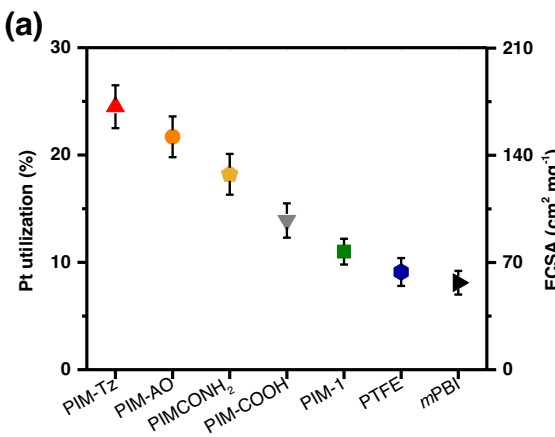

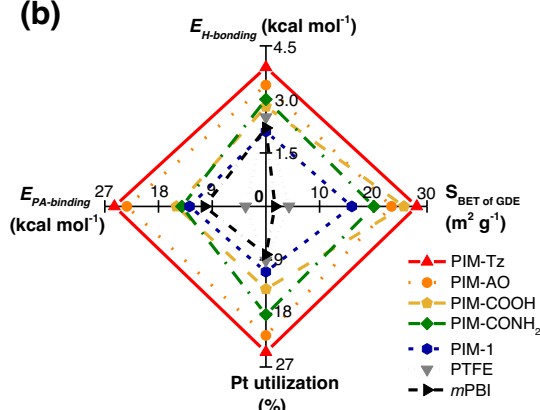

**Fig. 4 | Electrochemical surface area estimation of the MEAs using PIMs as binders in the catalyst layer. a** Pt utilization and ECSA of MEAs with different binder materials. **b** Correlation of Pt utilization with the hydrogen-bonding energy, PA binding energy and $S_{BET}$ of porous GDEs with different binders. The error bars indicate the standard deviation, calculated from the integral of the hydrogen desorption area of the CV curves. Test conditions are listed in Supplementary Table 1.

with amide, carboxylic acid and AO groups, indicating higher proton conduction in the catalyst layer with functional PIMs as binders. Due to the acidic nature of the functional groups in PIM-Tz and PIM-COOH, the acid-base interaction between these polymers and PA is weak while the hydrogen-bonding interaction may play a dominant role, thus leading to low ADL and good dimensional stability (swelling ratios < 8%), as seen from Fig. 1c, which are favorable to the cell device fabrication and thus performance being used as binders in the MEAs[8].

Subsequently, the effect of interaction with PA via acid/base interaction of the functional groups and hydrogen-bonding of the binder materials was further confirmed by the solid-state [31]P NMR analysis of the corresponding functional monomers with 1 equivalent PA (Fig. 2d and Supplementary Fig. 6) and also polymer powders saturated with PA (Fig. 2e). Generally, the acid-base interaction between functional groups and PA will result in chemical shifts appearing more downfield, while the hydrogen-bonding would induce the upfield chemical shifts[31]. As shown in Fig. 2d, the highest lowfield chemical shifts of [31]P were observed for PA doped benzimidazole (BIm) and benzylamidine (BAO) confirming their strongest acid-base interactions with PA[32]. On the contrary, the PA doped benzyltetrazole (BTz) showed the highest upfield chemical shift of [31]P indicating the strong hydrogen-bonding interaction[31,33]. When the polymer powders were saturated with pure PA, the same behaviors of chemical shifts of [31]P were observed in their [31]P NMR results. Although the hydrogen-bonds exist in the whole PA doped systems, PIM-Tz, PIM-AO and PIM-COOH showed higher upfield chemical shifts of [31]P in [31]P NMR, indicating the stronger hydrogen-bonding interactions with PA[34,35]. In case of PA doped PIM-1, PIM-CONH₂, and mPBI, the powders showed signals around 0 ppm, suggesting the weak hydrogen-bonding interactions between these polymers and PA. Moreover, the PA binding energy with repeat units of functional PIMs was calculated to be in the range of 12.8–25.4 kcal mol[-1] based on the hydrogen-bonding energy and acid-base interactions (Supplementary Fig. 3 and Data 1). As seen from Table 1, the PA binding energy of the functionalized PIMs having stronger hydrogen-bonds (e.g. higher hydrogen-bonding energy) are much higher than that of PTFE and mPBI. Most importantly, PIM-Tz exhibits a greater PA binding energy (25.4 kcal mol[-1]) than that of the Pt catalyst (19.5 kcal mol[-1]), implying the potential as a catalyst binder for preferential retention of PA in its vicinity and mitigation of the acid adsorption on Pt surface[1,23].

### Gas Diffusion Electrode (GDE) with functionalized PIM binders
The GDEs for HT-PEMFC with functional and microporous PIM binders were fabricated by a catalyst-coated substrate (CCS) process. The pore-size distribution of GDEs was determined by the quenched solid state

functional theory (QSSFT) equilibrium model. As seen from illustration in Fig. 3a the GDE with m-PBI binder exhibits a lower fraction of primary pores up to 20 nm than that of the GDE without binders. This behavior could be attributed to the dense structures of mPBI, which will fill up the primary pores when it is employed as binder material[8,36]. The binders of highly microporous PIMs increase the microporosity of GDE significantly in spite of the fact that they fill up the primary pores of catalyst layer, as shown in Fig. 3a. Although the PIMs with functional groups have lower BET values as discussed above, the higher microporosity of GDEs than that of the PIM-1 was observed mostly due to excellent dispersion of catalysts in the modified PIMs binder solution as confirmed by the SEM and Laser Scattering Particle Size Analyzer (LSPSA) results (Supplementary Figs. 7 and 8) resulting from their strong hydrogen-bonding between functional groups. Thus, the highest pore volume and BET values (Fig. 3b) were observed for the GDE having PIM-Tz as the binder. The large number of primary pores (1–20 nm) of PIM-Tz based GDEs would promote the transport of gases to the catalyst particles and the excess of product water, which will be discussed below[8].

In addition to the microporosity of GDEs, the interaction of GDE with PA was further characterized by the contact angle of PA, which is relevant to both the proton conductivity and mitigation of the PA flooding on the Pt surface in the catalyst layer. As shown in Fig. 3c, the surface of the mPBI bonded GDE was completely wetted by PA. This was also the case for the Pt/C GDE with no binder material, hinting the possibility of local flooding of PA on the catalyst surface in these two PA doped MEAs and ultimately results in low catalyst utilization. The introduction of functionalized PIMs into the CL increased the contact angles, thus mitigating the flooding of PA on the Pt surface. Unlike PIM-1 or PTFE bonded GDEs, which lack of the acidophilic groups, smaller PA contact angles were observed for the GDEs with functional PIM binders likely due to the hydrogen-bonding or/and acid-base interaction of the functional PIMs with PA. With high affinities to PA, the functional binder materials are expected not only to provide pathways for fast proton-conduction through the catalyst layer but also to avoid the local flooding of PA on the Pt surface, thus facilitating the establishment of triple-phase boundaries for the high electrochemical activity. Moreover, as shown in Supplementary Fig. 9, the contact angle of binder materials with water suggests the strong hydrogen-bonding of PIM-Tz with the ORR product, which is likely to enhance ORR reactivity[37].

### Electrochemical surface area estimation and fuel cell performance
The MEAs were assembled by sandwiching two electrodes with a PA-doped mPBI membrane and characterized in a single H₂-O₂ fuel cell.

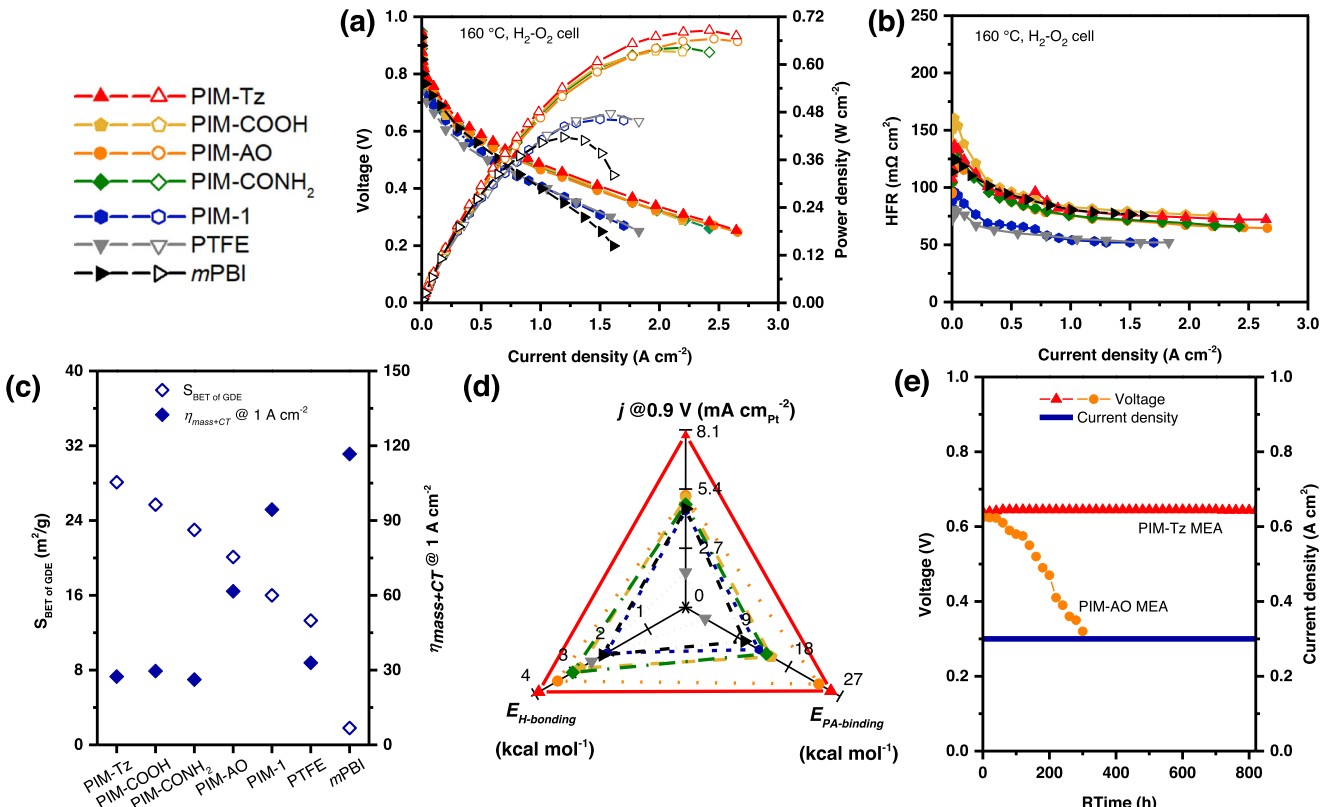

**Fig. 5 | Performance of the MEAs with PIMs binders in the catalyst layer without backpressure nor external humidification. a** Polarization (solid symbols) and power density (open symbols) curves, **b** high frequency resistance curves, and **c** the relation of the $S_{BET}$ of GDEs with the sum of mass loss and charger transfer loss of these GDEs at 1 A cm⁻². **d** The correlation of specific activity of the Pt catalyst at 0.9 V of the GDEs with hydrogen-bonding and PA binding energies of the binder materials. **e** Stability test of the single $H_2$-$O_2$ fuel cells with different binder materials. Test conditions are listed in Supplementary Table 1.

The anode and cathode contained $0.5\,mg_{Pt}\,cm^{-2}$ and 20 wt% binder materials in the catalyst layer. The electrochemical surface area (ECSA) was estimated by the cyclic voltammograms for the hydrogen desorption after about 24 h break-in period until the cell performance was stable (Supplementary Fig. 10)[11]. Based on the ECSA the Pt utilization is qualitatively compared by use of Eqs. 9 and 10 in the experimental part, as shown in Fig. 4a. Except the PIM-1 binder based GDE, a steady improvement of the Pt utilization is observed from mPBI to PIM-Tz based GDEs in the order of hydrogen-bonding energy values of the binder materials with 85% PA. Correlation of the Pt utilization results with the H-bonding energy, PA binding energy and the BET area of the GDEs with different binder materials is attempted, as shown in Fig. 4b. The BET was used as an indicator of the microporosity of the GDEs and hence the gas transport ability of the binder phase. The strong interfacial hydrogen-bonds in the functional binder materials show a significantly positive effect on the PA-bonding energy, and thus the Pt utilization in the MEAs in fuel cell devices. The binding energy between PA and functionalized PIMs is larger than the adsorption energy of PA on the Pt surface, resulting in the higher ECSA and enhanced Pt utilization, probably due to the favored sorption of PA on binder spots and the mitigation of the Pt poisoning[37]. Undoubtedly, as shown in Fig. 4b, the PIM-Tz with the highest hydrogen-bonding energy, PA binding energy, and microporous structure, shows the highest ECSA value of 182 cm² $mg_{Pt}^{-1}$ and hence better a Pt utilization than that of the MEAs with mPBI or PTFE binder in the catalyst layer (Fig. 4a).

Thus, the use of functional PIMs binder in catalyst layers leads to a substantial improvement in the $H_2$-$O_2$ fuel cell performance at low and high current densities compared to other binders. The peak power density and kinetic activity of these GDEs follow the order: PIM-Tz > PIM-AO > PIM-COOH > PIM-CONH₂ > mPBI > PIM-1 > PTFE    (Fig. 5a). Since the same type of mPBI/PA membranes was used in these MEAs,

the difference of high frequency resistance (HFR) (Fig. 5b) was recorded by electrochemical impedance spectroscopy (EIS). The HFR originates from the ohmic resistance of the membrane, which is determined by the amount of the doping acid. When a fuel cell is assembled and activated the doping acid transfers from the membrane to the catalyst layer[38]. The MEAs with binder materials having stronger affinity for PA show a large HFR, suggesting the fast transfer of the acid from the membrane to the catalyst layer, resulting in establishment of triple-phase boundaries and thus higher ORR activity compared to the MEAs with PIM-1 and PTFE as binders[11]. As a consequence such MEAs exhibit a relatively larger membrane resistance i.e. HFR. Moreover, the sum of the mass-transport and charge transfer losses ($\eta_{mass+CT}$) at 1.0 A cm⁻², as shown in Fig. 5c, further support this finding. For example, with the functional binder materials having stronger PA affinity, the MEAs showed less $\eta_{CT}$ than that for e.g. PIM-1 and PTFE binders (Supplementary Fig. 11). As a result, the highest fuel cell performance over the whole current density range, with the highest peak power density of 686 mW cm⁻² at the current density of 2.4 A cm⁻² was observed for the MEA having PIM-Tz as binder (Fig. 5a).

Moreover, the cell voltage at 0.3 A cm⁻² (Supplementary table 2) further confirmed the higher Pt utilization of the functional PIMs bonded electrodes. For example, taking a rated point at 0.3 A cm², the performance gain of the cell voltage is about 34 mV for PIM-Tz bonded electrodes in comparison to the m-PBI bonded electrodes. According to Eq. 14, using the average Tafel slope of 74 mV dec⁻¹ of these two electrodes (as shown in Supplementary Table 2), this voltage gain corresponds to an increase in the active Pt surface area by a factor of about 2.8 under assumption that the intrinsic ORR exchange current remains unchanged. This factor is consistent with the catalyst utilization discussed above. Thus, the highest specific area activity of Pt catalyst in PIM-Tz bonded electrodes at 0.9 V$_{iR\text{-free}}$ is estimated to be

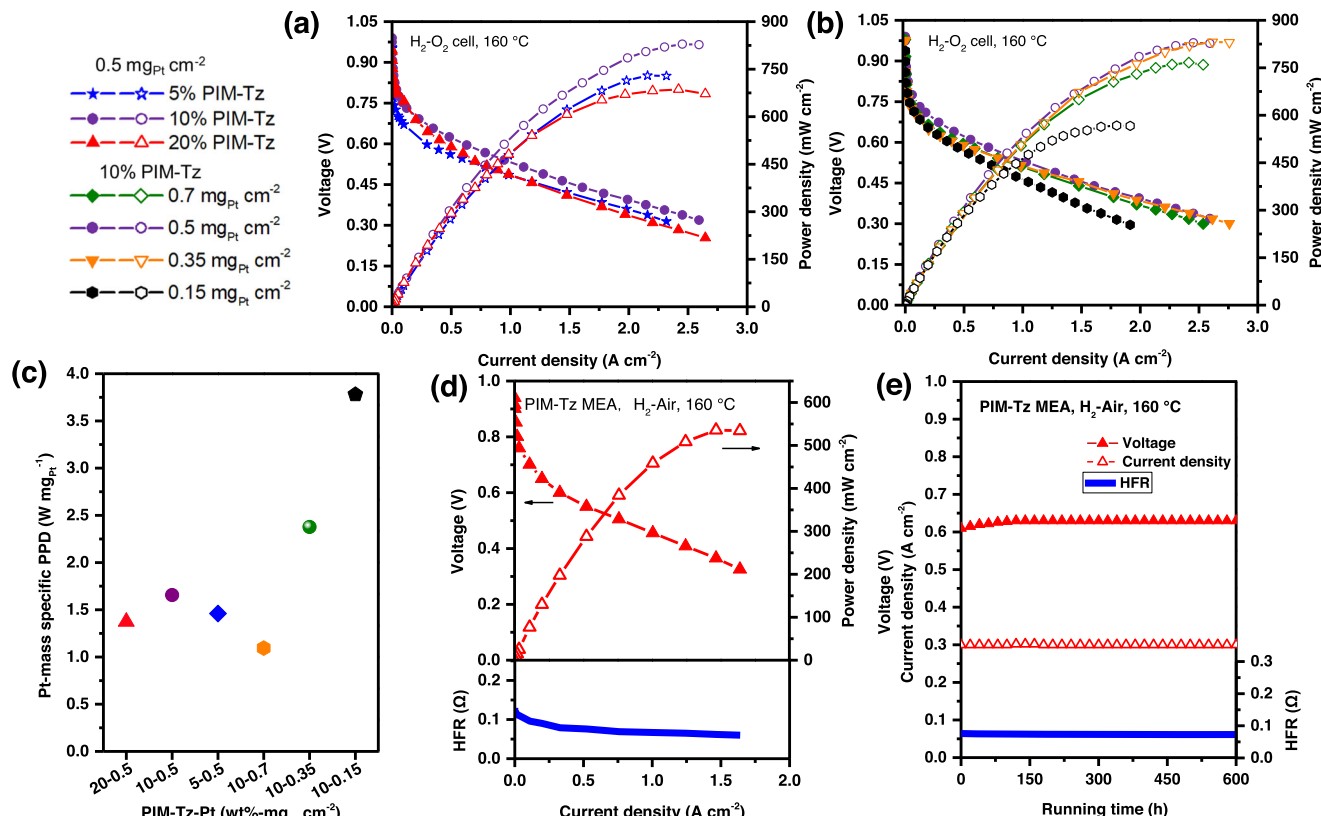

**Fig. 6 | HT-PEMFC performance of the MEAs using PIM-Tz as binder in the catalyst layer. a** Varied PIM-Tz contents in the catalyst layer with the same Pt catalyst loading 0.5 mg$_{Pt}$ cm$^{-2}$ and (**b**) varied Pt loading with a constant (10 wt%) PIM-Tz content. Solid symbols for the voltage and open symbols for the power density. **c** Pt-mass specific peak power density for combinations of the binder contents and Pt loadings. **d** Performance and (**e**) stability at current density of 0.3 A cm$^{-2}$ of PIM-Tz binder based H$_2$-Air fuel cell. The test conditions have been listed in Supplementary Table 1.

7.9 μA cm$_{Pt}^{-2}$, is about twice that in *m*-PBI bonded electrodes[39]. It seems safe to conclude that the specific area activity of Pt catalysts at 0.9 V$_{iR-free}$ correlates well with the catalyst binder materials using descriptors of the hydrogen-bonding and PA binding energy as shown in Fig. 4d. Furthermore, the single H$_2$-O$_2$ cell tests were further extended to a period of 800 h under a constant current density of 0.3 A cm$^{-2}$. Though the presence of PBI as the catalyst binder is reported to influence the Pt dissolution during the fuel cell operation[40], the electrochemical stability of the tetrazole groups of the PIM binders does not seem an issue[41,42]. A stable cell voltage of 0.640 V was observed during the test when a slight increase of 5 mV was in fact measured. The results are very encouraging to achieve the target long-term durability as listed in Supplementary Table 2[42]. The PIM-AO based MEAs, however, showed a faster cell voltage decay rate, probably due to the excessive dimensional swelling of the binder upon the PA uptake.

## Low Pt loading in fuel cells

Effort was made to optimize the amount of the PIM-Tz binder in the catalyst layer. At a constant Pt loading of 0.5 mg$_{Pt}$ cm$^{-2}$ the optimal binder content was found to be 10% PIM-Tz in the studied range from 5 to 20 wt%. The hypothesis was that a higher binder loading would result in a thicker coating layer onto the catalyst particles and increase the mass transport resistance, while a lower binder loading reduces results in reduced catalyst coverage. As shown in Fig. 6a, the highest cell performance was obtained for the MEAs with a binder concentration of 10% and catalyst loading of 0.5 mg$_{Pt}$ cm$^{-2}$. Considering the use of the same catalyst of Pt/C and microporosity of binder, the Pt utilization increase is more plausibly explained by the stronger hydrogen-bonding/higher PA binding energy leading to maximizing the activity of Pt and better mass transport. More interestingly, the

electrode performance could be further improved by lowering the catalyst loading at the fixed binder loading of 10% (Fig. 6b). At a Pt loading of as low as 0.15 mg$_{Pt}$ cm$^{-2}$ in the catalyst layer, for example, the high catalyst utilization of 57% was achieved (Supplementary Fig. 12b), corresponding to a Pt-mass specific power density of 3.8 W mg$_{Pt}^{-1}$ at 160 °C. These are highly competitive to that of 0.1–3.0 W mg$_{Pt}^{-1}$ for MEAs with the catalyst loading of <0.7 mg$_{Pt}$ cm$^{-2}$, as reported in literatures (Supplementary Table 3)[8,10,11,43–47]. The Pt-mass specific power density was found to be 2.4 W mg$_{Pt}^{-1}$ at the Pt loading of 0.35 mg$_{Pt}$ cm$^{-2}$ (Fig. 6c).

Furthermore, the PIM-Tz based MEAs were tested in the H$_2$-air mode at the fixed binder loading of 10% (Fig. 6d). Although the catalyst loading was as low as 0.35 mg$_{Pt}$ cm$^{-2}$, a significant peak power density reached 536 mW cm$^{-2}$ at the operating temperature of 160 °C, suggesting high ORR activity and good mass transport at the PIM-Tz bonded electrode. Moreover, as shown in Supplementary Table 3, the performance demonstrated here with PIM-Tz was substantially higher than that of the reported HT-PEMFC using similar or even higher Pt loadings. Especially, as shown in Supplementary Fig. 13, at a loading of 0.15 mg$_{Pt}$ cm$^{-2}$, a peak power densities of H$_2$-Air fuel cell exceeding 360 mW cm$^{-2}$ was obtained, which is on par with state-of-the-art cells with several times higher Pt loading (Supplementary Table 4). Finally, the performance stability was preliminarily evaluated at a constant current density of 0.3 A cm$^{-2}$ and 160 °C. As shown in Fig. 6e, for the MEA with the catalyst loading of 0.35 mg$_{Pt}$ cm$^{-2}$, the cell voltage increased from 0.62 to 0.63 V during the first 150 h due to the electrode break-in process with PA redistribution[37]. The cell performance remained stable for >500 h with no visible voltage decay, indicating the technical feasibility of the PIM-Tz binder material for the HT PEMFC.

Overall, in the present work, a series of polymers with intrinsic microporosity (PIM) are modified by introduction of functional groups. The polymers have gas permeabilities of two orders of magnitude higher than PTFE and *m*PBI, the conventional catalyst binders. The introduced functional groups possess varied p$K_a$ values and capabilities of hydrogen-bonding/acid-base interactions. The interaction of the binder polymers with PA is characterized in terms of the PA binding energy and the hydrogen-bonding energy and studied by theoretical calculation, acid doping/swelling and NMR spectra. The tetrazole-functionalized polymer (PIM-Tz) showed the optimal characteristics and particularly the higher PA binding energy than the adsorption energy of PA on Pt particles. This makes PIM-Tz an ideal binder for preferential retention of PA in the catalyst layer, which ensures the ionic conductivity and, at the same time, mitigates the acid flooding and adsorption of the Pt particles. Combination of the intrinsic microporosity and the enhanced PA binding energy improves the gas diffusion electrodes in terms of extended triphase boundaries, mass transportation and hence the overall fuel cell performance. Optimized electrodes are manufactured using 10 wt% PIM-Tz in the catalyst layer with a platinum loading of 0.35 mg$_{Pt}$ cm$^{-2}$. Fuel cell tests under H$_2$-O$_2$ and H$_2$-air operation show a peak power density of 832 mW cm$^{-2}$ and 536 mW cm$^{-2}$, respectively. Most importantly, with the low Pt loading of only 0.15 mg$_{Pt}$ cm$^2$, the highest Pt-mass specific power density of as high as 3.8 W mg$_{Pt}^{-1}$ for H$_2$-O$_2$ cell, and of 2.4 W mg$_{Pt}^{-1}$ for H$_2$-air cell were achieved. These excellent performances are highly competitive to that of reported in literatures. Therefore, the application of hydrogen-bonding dependent PIM-Tz with high PA binding energy as binder material affords great opportunities to lower the catalyst loading and provide a viable cost-saving alternative for commercial HT-PEMFCs.

## Methods

### Materials

*N*-methylpyrrolidone (NMP, 99.5%), *N,N*-dimethylacetamide (DMAc, 99.5%), *N,N*-Dimethylformamide (DMF, 99.5%), toluene (99.0%), methanol (CH$_3$OH, 99.0%), 1,4-dioxane, ethanol (CH$_3$CH$_2$OH, 99.0%), acetic acid (CH$_3$COOH, 99.5%), sodium azide (NaN$_3$, 99.0%), zinc chloride (ZnCl$_2$, 99.95%), sodium hydroxide (NaOH, 99.5%), sulfuric acid (H$_2$SO$_4$, 96%), hydrogen peroxide (H$_2$O$_2$, 25 wt% solution), potassium carbonate (K$_2$CO$_3$, 99.0%), sodium bicarbonate (NaHCO$_3$, 99.0%), dimethyl sulfoxide (DMSO, 99.5%), hydroxyl amine (50 wt% solution in water, 99.999%), tetrahydrofuran (THF, 99.5%), chloroform (CHCl$_3$, 99.5%), isopropanol (99.5%), CDCl$_3$ (99.8%), DMSO-$d_6$ (99.96%) and methyl orange (99.5%) were purchased from Shanghai Aladdin Biochemical Technology Co., LTD and used as received. 5,5',6,6'-Tetrahydroxy-3,3,3',3'-tetramethylspirobisindane (TTSBI > 96%, Aladdin Biochemical Technology Co., LTD) was purified by washing with CH$_3$COOH. Tetrafluoroterephthalonitrile (TFTPN > 98%, Shanghai Aladdin Biochemical Technology Co., LTD) was purified and K$_2$CO$_3$ was dried by vacuum sublimation at 150 °C under dry atmosphere. *N,N*-Dimethylformamide (DMF) was dried by activated molecular sieves and used freshly after distillation under a N$_2$ atmosphere. Prior to use, acetone was distilled over CaSO$_4$ under a N$_2$ atmosphere. The 85.0% phosphoric acid (PA) was purchased from Tianjin Fengchuan Fine Chemicals Co., Ltd. Pt/C (HPT040, 38.90–41.10 wt% Pt, particle size < 4.5 nm), Teflon® PTFE DISP 30LX fluoroplastic aqueous dispersion (60% wt, average particle size: 0.230 μm, produced by DuPont) and carbon paper with gas diffusion layer (GDL) (HCP120, HESEN, China) were purchased from Shanghai Hesen Electric Appliance Co., LTD. Commercial GDE with the catalyst loading of 0.90 mg$_{Pt}$ cm$^{-2}$ was obtained from Blue World Technologies.

### Synthesis of PIM-1 and its derivatives

PIM-1 and the four derivatives thereof with different hydrophilic groups (i.e. amide-PIM-1 (PIM-CONH$_2$), amidoxime-PIM-1 (PIM-Mi),

carboxylated-PIM (PIM-COOH) and tetrazole-PIM-1 (PIM-Tz)) were prepared according to the literature procedures (Supplementary Fig. 2). The number-average molecular weight ($M_n$) was determined by gel permeation chromatography (GPC) on an instrument equipped with a Waters 1515 isocratic HPLC pump and Waters 2414 refractive index detector. It was calibrated by polystyrene standards, using HPLC-grade tetrahydrofuran as mobile phase.

**PIM-1.** PIM-1 was prepared by polycondensation polymerization of TTSBI and TFTPN according to procedure reported by Budd et al.[21]. In a typical procedure, 3.0 g (15.0 mmol) of TFTPN, 5.1 g (15.0 mmol) of TTSBI, 105 mL of anhydrous DMAc and 6.2 g of K$_2$CO$_3$ were added into a three-neck round bottom flask and then stirred under N$_2$ atmosphere at 155 °C for about 15 min. Thereafter, the mixture was poured into methanol and generating precipitate was collected by filtration. The crude product was dissolved in chloroform and re-precipitated by adding methanol. Luminous yellow solids of PIM-1 were obtained by drying the crude product in an oven at 110 °C overnight (yield: 5.0 g, 61.3 %, $M_n$ = 70 kDa).

**PIM-CONH$_2$.** PIM-CONH$_2$ was synthesized by post-synthesis modification method of PIM-1 using hydrogen peroxide (H$_2$O$_2$) as reagent at room temperature[28]. PIM-1 (1.0 g, 2.2 mmol repeat units) and 60 mL (844.8 mmol) of DMSO were added into a 250 mL a three-necked round bottomed flask and stirred for 1 h at 20 °C. 1.5 g (10.8 mmol) of K$_2$CO$_3$ was added into the above solution until the pH value of the resulting mixture reached between 9-10. Then 10.0 mL of 25 wt% H$_2$O$_2$ (65.3 mmol) was added in drops. The reaction mixture was stirred at 20 °C for 24 h. After completion of the reaction, the mixture was poured into 500 mL water and stirred at 20 °C overnight. The yielded yellow solid was collected by vacuum filtration, washed with water and methanol, and then dried in a vacuum oven at 25 °C for 24 h. The final product was a milky yellow, free-flowing powder with 100% conversion degree (yield: 0.8 g, 80.0%, $M_n$ = 42 kDa).

**PIM-AO.** PIM-AO was synthesized by modifying PIM-1 polymer following a published protocol[48]. Under a N$_2$ atmosphere, a solution of 0.6 g of PIM-1 powder (1.4 mmol repeat units) in 40 mL THF was added into a three-necked round bottomed flask equipped with condenser and thermometer, then stirred and heated. When the temperature reached 65 °C, 6.0 mL hydroxyl amine was added in drops using a syringe, and the resulting mixture was refluxed at 69 °C for 20 h. When the reaction was complete, the resulting hazy solution was cooled to 20 °C and 150 mL ethanol was added to precipitate the white solid. The precipitates were filtered and washed with 50 mL ethanol four times, then dried at 110 °C for 3 h. The final product of PIM-AO with 100% conversion degree was obtained as an off-white powder (yield: 0.5 g, 83.3%, $M_n$ = 53 kDa).

**PIM-COOH.** PIM-COOH was synthesized through acidification of PIM-1[49]. Under N$_2$ atmosphere, the miscible liquids of 1.85 g PIM-1 (4.3 mmol repeat units) in 15 mL concentrated H$_2$SO$_4$, 15 mL H$_2$O and 10 mL glacial acetic acid, were added into a round bottomed flask, stirred and heated at 105 °C for 48 h. Then the mixture was cooled and diluted with 200 mL deionized water. The resulting precipitate was collected by filtration, and rinsed with water and methanol, then dried in a vacuum oven at 50 °C for 24 h to give the product with 100% conversion degree appeared as a fluorescent, free-flowing powder (Yield: 1.3 g, 65.0%, $M_n$ = 40 kDa).

**PIM-Tz.** PIM-Tz was synthesized by [2+3] cycloaddition reaction under N$_2$ atmosphere[29]. A solution of 1.1 g PIM-1 (2.5 mmol repeat units) in 50 mL DMAc was introduced into a flask containing NaN$_3$ (1.3 g, 20.0 mmol) and anhydrous ZnCl$_2$ (2.7 g, 20.0 mmol). According to the ref. 29., the reaction mixture was stirred and retained at 120 °C for

about 2.5 days, then cooled and poured into 50 mL of 1 M HCl. The resulting mixture was heated at 60 °C for 1 h, filtered, and the precipitate was washed with diluted HCl (36% wt HCl: $H_2O$ = 1:50 (V: V)), $H_2O$ and acetone in turn. The product was obtained with about 75% conversion degree after being dried in a vacuum oven at 120 °C for 24 h (yield: 1.2 g, 90.0%, $M_n$ = 41 kDa).

## Preparation of PA doped mPBI and PIM membranes

The mPBI was synthesized according to ref. 46. ($M_w \approx$ 50 kDa, inherent viscosities: 0.88 dL g$^{-1}$ at 30 °C in 96% $H_2SO_4$ (0.5 g/dL)) and dissolved in NMP to form a solution of 5 wt% concentration. Polymer films were prepared by solution casting after filtering and dried at 80 °C for 12 h in a vacuum oven. The obtained film was then boiled in deionized water for 12 h to remove residual NMP and dried again. The obtained films with a thickness about 40 μm were immersed in an 85.0% PA solution at 20 °C for 2 days until their weight reached a constant value, then wiped used filter papers. The PA-doping level (ADL) was about 10.5 mol PA per mol repeat unit, as determined by acid-base titration using methyl orange as an indicator[50]. The resulting mPBI/PA had a thickness of about 80 μm the specific conductivity was ~90 mS cm$^{-1}$ at 160 °C. The solution of as-synthesized PIM-1 in $CHCl_3$ (5 wt%) and the hydropilicized PIMs in NMP (5%) were prepared at room temperature. The degassed solution was cast on a glass plate and dried at room temperature or in an oven at 80 °C for 24 h. The obtained membranes were delaminated from the plate by immersing in deionized water and boiled in deionized water for 12 h to remove trace amounts of NMP, and then dried in a vacuum oven at 80 °C for 24 h. The thickness of the obtained membranes was 30 μm.

## Preparation of GDEs

The membrane electrode assemblies (MEAs) for fuel cell testing were prepared by catalyst-coated substrate (CCS) method[11,18]. The thickness of the resulting GDE was about 300 μm. The catalyst loading of the final GDEs was 0.70–0.15 mg$_{Pt}$ cm$^{-2}$ and binder content was 5–20 wt% (in relation to the solid content in the catalyst ink).

**PTFE electrode**. 30.0 mg Pt/C was first wetted by 225.0 mg of distilled water and thereafter 7.5 mg of PTFE and 1.35 g isopropanol were added in turn. The resulting mixture was stirred for 30 min and then ultrasonicated for 60 min to form a catalyst ink. The ink was sprayed onto carbon paper with GDL using a spray gun. The PTFE GDE was obtained by sintering at 350 °C for 30 min in $N_2$ atmosphere using tubular furnace.

**PIMs GDE**. The GDEs containing the PIM binders were prepared without sintering by a similar procedure as the PTFE electrode. A homogeneous solution of 7.5 mg PIM in 1.35 g DMAc was added to a mixture of 30.0 mg Pt/C and 225.0 mg of water. The resulting mixture was stirred for 30 min and sonicated for 60 min. Instead of brushing, the slurry was sprayed onto carbon paper with GDL using a spray gun, and the prepared GDEs were obtained after being kept at 150 °C for 1 h to evaporate DMAc.

## PA Doping

The PA-doped samples were obtained by immersing PIM films in 85% PA solution at 60 °C for 8–24 h until their weight reached a constant value, followed by wiping with filter papers. The PA doping level and the thickness expansion of the films were recorded and calculated as the average and standard deviation for three independent samples. The value of PA uptatke and ADL were determined by acid-base titration using methyl orange as an indicator[41]. The PA-doped membranes were cut into 1 cm × 4 cm size and these samples were immersed in DI water for 2 h. The solutions were titrated with 0.1 M NaOH statndard solution, respectively. After neutralization, the samples were taken out and thoroughly washed with DI water. The samples were weighed

($W_{dry}$) after being dried in a vacuum oven at 100 °C for 4 h. The PA uptake was calculated according to the Eq. 1, where $C_{NaOH}$ and $V_{NaOH}$ are the molar concentration and volume of standard NaOH solution respectively. $Equiv_{mol}$ is the equivalent mole of titrant for PA (in this case $Equiv_{mol}$ = 1). $W_{dry}$ is the weight of dry sample and 98.0 (g mol$^{-1}$) represents the molecular weight of $H_3PO_4$.

$$\text{PA uptake (\%)} = \frac{V_{NaOH} \times C_{NaOH} \times 98.0}{Equiv_{mol} \times W_{dry}} \times 100\% \quad (1)$$

The acid doping level (ADL) was calculated according to Eq. 2, where used $MW$ (g mol$^{-1}$) is the molecular weight of the polymer repeat unit.

$$\text{ADL} = \frac{V_{NaOH} \times C_{NaOH} \times MW}{Equiv_{mol} \times W_{dry}} \quad (2)$$

The thickness of the membranes was measured before and after PA doping. Then, the swelling ratios ($SR_T$) were calculated according to Equation 3, where $T_{dry}$ and $T_{wet}$ are the thickness of membranes before and after PA doping, respectively.

$$SR_T = \frac{T_{wet} - T_{dry}}{T_{dry}} \times 100\% \quad (3)$$

## Characterization and measurements

**Calculation of binding energy with phosphoric acid.** All the theoretical calculations for molecular geometry optimizations were performed in Materials Studio software package Dmol[51,52]. The GGA-BLYP functional and double numerical plus polarization basis set were employed for the calculations[53]. Conductor-like screening model (COSMO) was introduced to solvent effects using water, ε = 78.54. Forcite plus module was performed to obtain the final energies of molecular structure for analysis, and the forcefield was based on Dreiding in the whole simulations. Binding energies were calculated as shown in Eq. 4, where $E_{Polymer}$ is the energy of the polymer chain, $E_{(Polymer + Phosphoric\ acid)}$ is the energy of the polymer chain and phosphoric acid molecule, and $E_{Phosphoric\ acid}$ is the energy of the phosphoric acid molecule.

$$\Delta E_{binding\ energy} = E_{Polymer} + E_{Phosphoric\ acid} - E_{(Polymer + Phosphoric\ acid)} \quad (4)$$

Materials Studio (version 8.0) software was employed for the theoretical calculations. The molecular dynamic (MD) simulations were performed by the Forcite module to obtain the structures in dynamical equilibration[54]. The forcefield was based on Dreiding[55]. The Andersen and Berendsen algorithms were used to control the temperature with a collision ratio of 1 and the pressure with a decay constant of 0.1 ps[56,57]. After the trajectory converges at a specific temperature, the process following 100 ps quench dynamics calculation at 1 atm, 100 ps NPT-MD simulation of at 1 atm, 50 ps NPP-MD simulation at 1 GPa, 20 ps NVT-MD simulation and 50 ps NPT simulation at 1 atm were carried out successively. Finally, the equilibration of the 3D amorphous model was conducted by NVT simulation for 5 ns.

The electronic structure calculations were performed for the adsorption energy calculations using CASTEP (Cambridge serial total energy package) program module[58]. The exchange correlation functional used was GGA-PBE[59,60]. Extended Pt(111) was modeled using supercells with the dimensions of 22.19 Å × 21.79 Å and the surface was modeled using four layers of metal atoms. Additional vacuum layer of 20 Å is added to avoid the periodic interactions, resulting in a unit cell with 272 metal atoms. A plane wave cutoff energy of 321 eV was used in all calculations. The convergence tolerances for geometry optimization calculations were set to the maximum displacement of 0.002 Å,

the maximum force of 0.03 eV Å$^{-1}$, the maximum energy change of $1.0 \times 10^{-5}$ eV atom$^{-1}$ and the maximum stress of 0.05 GPa. Adsorption energies were calculated using the Eq. 5, where $E_{Pt}$ is the energy of the Pt surface, $E_{(Pt+H_2PO_4^-)}$ is the energy of adsorbed on the Pt(111) surface, and $E_{H_2PO_4^-}$ were the energy of $H_2PO_4^-$ in the unit cell of 20.00 Å × 20.00 Å × 20.00 Å.

$$\Delta E_{\mathrm{adsorption}} = E_{H_2PO_4^-} + E_{\mathrm{Pt}} - \mathrm{E}_{(Pt+H_2PO_4^-)} \tag{5}$$

Predicting and estimating p$K_a$ values with the Relative Gibbs free energy change (RGC) approaches and blinding energy using density functional theory are as follow: Dmol$^3$ was used for molecular geometry optimizations[51,52]. The GGA-BLYP functional and double numerical plus polarization basis set were employed for the calculations[53]. A 3.7 Å real space cutoff was employed as atomic orbital. A threshold value of $10^{-6}$ Ha was converged for spin-restricted self-consistent field calculations. Custom convergence criteria was specified as $5 \times 10^{-6}$ Ha for energies, $1 \times 10^{-3}$ Ha/Å for gradient, and $5 \times 10^{-3}$ Å for displacement in entire computational procedure. Solvent effect was included by the usage of a conductor-like screening model (COSMO) (water, $\varepsilon = 78.54$). All single molecules were fully optimized to ensure there is no imaginary frequency. The p$K_a$ values of the compound were calculated using Eq. 6, where $pK_a(S_1)$ and $pK_a(S_2)$ are the $pKa$ value of the reference acid, $= \frac{\Delta G - \Delta G(S_1)}{\Delta G(S_2) - \Delta G(S_1)}$, and $\Delta G$ is the Gibbs free energy difference between the acid and its conjugate base.

$$pK_a = [pK_a(S_1) - pK_a(S_2)]RGC + pK_a(S_1) \tag{6}$$

**Gas permeation tests of PIMs.** The H$_2$ and O$_2$ permeability of membranes was measured by the constant-volume permeation cell using same film preparation protocols according to our previous work[61]. The average thickness was taken by averaging the thickness measured at three points of the membranes and the effective area is 0.5 cm$^2$. The permeation cell after mounting the membrane was first evacuated under vacuum for at least 12 h. By using the constant-pressure/variable-volume method, to ensure the accuracy and reproducibility, permeability coefficients ($P$) of O$_2$ and H$_2$ were obtained by averaging the results from at least three-time tests at 35 °C with a feed pressure of 0.4 MPa and atmospheric permeate pressure. A mass flow controller (Agilent ADM2000) was used to control the permeation flow. The upstream pressure ($P_0$) was maintained constant. A MKS Baratron® pressure transducer was used to measure the increasement of downstream pressure ($p$) in the constant volume reservoir ($V$). The gas permeability ($P$) is obtained from the slope (d$p$/d$t$) at steady state in the curve of downstream pressure ($p$) over time according to Eq. 7, where $P$ is the gas permeability in Barrer (1 Barrer = $1 \times 10^{-10}$ cm$^3$(STP) cm/cm$^2$ s cmHg), $V$ is the volume of the downstream reservoir (cm$^3$), $A$ is the effective area of membrane (cm$^2$), $L$ is the thickness of the membrane (cm), and $T$ is the temperature (K).

$$P = \left(\frac{273 \times 10^{10}}{760}\right)\left(\frac{VL}{ATP_0 \times \frac{76}{14.7}}\right)\left(\frac{dp}{dt}\right) \tag{7}$$

**Proton conductivity.** The proton conductivity ($\sigma$) of the PA doped membranes was measured by electrochemical impedance spectroscopy (EIS) on a Bio-Logic VSP-300 with a frequency ranging from 1.0 MHz to 10.0 Hz. The testing cell mounted with membrane was placed in a chamber with controlled temperature. The conductivity was calculated from Eq. 8 in which $A$ is the cross-sectional area of sample and $L$ is the distance between two electrodes of the cell; $R$ is the recorded resistance on the Nyquist curve.

$$\sigma = \frac{L}{A \times R} \tag{8}$$

**$^{31}$P NMR analysis.** Solid state $^{31}$P NMR spectra of the PA doped samples were acquired on Bruker Avance 600 MHz Wide Bore spectrometer (14.1 T) using 4 mm HXY probe with ZrO$_2$ rotor, DR mode, lambda/2 and range coil. MAS spinning rate was 9 kHz and chemical shifts were referenced relative to (NH$_4$)$_2$HPO$_4$, 1.00 ppm. A certain amount of pure PA was well ground with polymer powder in a glove box under an inert atmosphere of Ar. The amount of pure PA added in the polymer powders is equivalent to the molar PA absorbed in the membranes, while the amount of pure PA added in the monomer powders is about 1 equivalent to the monomers.

**Micro-porosity characterization.** N$_2$ adsorption/desorption of all polymers and the GDEs were measured using a Micromeritics TriStar II 3020 3.02. The samples (polymer powder and GDEs) were degassed at 393 K for 12–16 h under vacuum before analysis. The time to adsorption equilibrium was about 180 s and the consecutive pressure value was within $1.3 \times 10^{-4}$ bar during the equilibration time. The specific surface areas were calculated by the theory of Brunauer, Emmett, and Teller (BET) from the N$_2$ adsorption isotherm (S$_{BET}$). The pore-size distribution is obtained by the quenched solid state functional theory (QSSFT) equilibrium model.

**Surface morphology of the catalyst layer.** The surface morphology of the catalyst layers was investigated using a scanning electron microscope (SEM, JSM-7900F) with an acceleration voltage of 20 kV.

### Laser scattering particle size analyzer

The particle size and distribution of the catalyst ink was measured by a Laser Scattering Particle Size Analyzer machine Beckman Coulter LS13320. The preparation of catalyst ink for Laser Scattering Particle Size Analyzer was the same as that prepared for GDEs, and the ink was diluted 5 times with DMAc or NMP. Ethanol was used as the flowing phase.

### Contact angle measurements

GDE samples with an area of 1 cm$^2$ were dried overnight in a sealed desiccator for contact angle measurement. The contact angles were measured using a manual baseline by a contact angle goniometer (SL-200B) with deionized water (drop needle diameter 5.6 mm). Equilibrium sessile drop contact angles were determined from the steady-state angles, which were typically observed to reach a constant value between 10 and 60 s after the drop contacted the GDE surface. Contact angle titrations were performed by measuring no less than three sets of contact angles 5 μL drops. Sessile drop contact angles were measured as the angle between the baseline of a liquid drop and the tangent at the solid–liquid boundary.

### Electrochemical measurements and analysis

CVs was measured at 160 °C with H$_2$ flow of 100 sccm and N$_2$ flow of 200 sccm through the anode and cathode respectively. The CVs were conducted between 0.08 V and 1.20 V after purging dry N$_2$ for 10 min at the flow rate of 200 sccm through the cathode at a scan rate of 50 mV/s. The electrochemical surface area (ECSA) was calculated based on hydrogen desorption data according to the Eq. (9). The Pt utilization at the cathode was calculated by normalizing the ECSA value with the physical surface area (700.9 cm$^2$ mg$_{Pt}^{-1}$) of the 40% Pt/C catalyst (d$_{Pt}$ = 4 nm), as shown in Eq. 10.

$$ESCA(cm^2 mg_{Pt}^{-1}) = \frac{Chargearea(mA\ cm^{-2}\ V)}{0.21 \times 10^{-3}(C\ cm_{Pt}^{-2}) \times Scanrate(mV\ S^{-1}) \times loading_{Pt}(mg\ cm^{-2})} \tag{9}$$

$$Pt\ utilization = \frac{ECSA}{700.9} \times 100\% \tag{10}$$

The MEA with an area of 4.00 cm² was prepared by sandwiching the PA/PBI membranes between two GDEs and hot-pressed at 100 °C under the pressure of 0.5 MPa for 5 min to eliminate the danger of insufficient compatibility of membrane material and GDEs. For each tested cell, anode and cathode used the same electrodes. A fuel cell workstation (Smart 2-WonATech Inc., Korea) was employed to check the repeatability of the cell performance. A Bio-Logic VSP-300 potentiostat was used to measure the electrochemical impedance spectra, cyclic voltammograms and $H_2$ cross-over current. Dry $H_2$ and $O_2$ were flown through the anode and cathode at 200 sccm, respectively. The fuel cell was tested at 160 °C with no backpressure, respectively. The MEAs were activated at a constant voltage of 0.3 A until the current became stable. Then steady-state polarization curves were recorded by polarizing the cell voltage from 1.0 V to 0.15 V in steps of 0.05 V, and holding the voltage for 2 min at each point. The high frequency impedance (HFR) measurement at 1 kHz was conducted when the cell reached steady state at various cell voltages using Bio-Logic VSP-300 potentiostat. For each binder, three MEAs were fabricated and tested to ensure the reproducibility of the data. The normalized standard deviations of the data were estimated to be <5%. Electrochemical impedance spectra were collected using the impedance analyzer in galvanostatic mode at different current densities with the frequency ranged from 10 kHz to 1 Hz and peak-to-peak perturbation of 5 mV.

For $H_2$-air cell test, the area of the MEA was 1.00 cm², a VersaStat 3 potentiostat was used to measure the electrochemical impedance spectra, cyclic voltammograms and cell performance. The flow of the dry $H_2$ and $O_2$ was 30 and 100 sccm respectively.

$$Pt - mass\,specific\,power\,density\,(W\,mg_{Pt}^{-1}) =$$
$$\frac{Peak\,Power\,density\,(W\,cm^{-2})}{Pt\,loading\,of\,the\,cathode\,(mg_{Pt}\,cm^{-2})} \quad (11)$$

The theoretical analysis of I-V curves and the electrochemical characteristics of the cells are quantified the voltage loss contributions from the various electrochemical processes, such as activation overpotential ($\eta_{act}$), ohmic over potential ($\eta_{ohm}$) and mass transfer overpotential ($\eta_{mass}$), which help to understand how the design of using PIMs binders with hydrophilic groups affects the cell performance more profoundly. Due to the greater exchange current density of hydrogen oxidation reaction (HOR) on anode, both kinetic and mass transfer overpotential for HOR were negligible. Therefore, the cell voltage, $E_{cell}$, of a $H_2/O_2$ fuel cell can be expressed using Eq. 12.

$$E_{cell} = E_{rev(P_{H_2}, P_{O_2}, T)} - \eta_{act} - \eta_{ohm} - \eta_{mass} \quad (12)$$

$$E_{rev(P_{H_2} P_{O_2} T)} = -\frac{\Delta H - T\Delta S}{nF} + \frac{RT}{nF} \ln \frac{P_{H_2} P_{O_2}^{0.5}}{P_{H_2O}} \quad (13)$$

The activation overpotential, $\eta_{act}$ was extrapolated from the Tafel plots. The ohmic overpotential, $\eta_{ohm}$, at 0.3 A cm⁻² was determined using EIS. The mass transport loss, $\eta_{mass}$, was calculated according to Eq. 5. The reversible potential of $H_2$-$O_2$ fuel cell $E_{rev(P_{H_2} P_{O_2} T)}$ depends on the cell temperature and the partial pressures of the reactants which can be calculated from Nernst equation (Eq. 13)) and is equating to 1.139 V under the operating conditions at 160 °C with no back pressure. Moreover, at a rated point in the I-V curve, the voltage of the cell can be calculated as follows:[62]

$$E_{cell} = -b \log \frac{i}{i_{GDE}^0} \quad (14)$$

where $b$ is the Tafel slope, $i$ (A cm⁻²) represents the current density at a rated point, e.g. i = 0.3 A cm⁻², $i_{GDE}^0$ (A cm$_{Pt}$⁻²) is the exchange current density (kinetic rate of ORR) of Pt catalyst in the electrode.

Considering the use of the same catalyst of Pt/C in the electrodes, $i_{Pt}^0$ for the GDEs without binders will be the same. Therefore, $i_{GDE}^0$ correlates well with the active electrochemical surface area with different binder materials in the catalyst layer, thus leading to different voltages at a rated point in the I-V curves.

## Data availability
The authors declare that the data supporting the findings of this study are available within the paper, Supplementary Information and Supplementary Data consisting the coodinates for the calculation of PA interaction with the binder materials and binder materials' pKa value. Further data beyond the immediate results presented here are available from the corresponding authors upon reasonable request.

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

## Acknowledgements

We thank Y. Li from Soochow University (computational chemistry materials simulation and design) for all the theoretical calculations for molecular geometry optimizations. We are grateful to financially supported by the National Natural Science Foundation of China (No. 21835005 and No. 52G15023) and China Scholarship Council. We also acknowledge the financial support from Innovation Fund Project of Shanxi Institute of Coal Chemistry, Chinese Academy of Sciences (SCJC-HN-2022–16), the autonomous research project of SKLCC (Grant No.: 2020BWZ001) and the Danish EUDP program (COBRA-Drive).

## Author contributions

H.T. and N.L. developed the concept. H.T. designed the experiments, performed the polymer characterization, MEA fabrication, testing experiments of fuel cells and data collection. K.G., Q. J., D.A., J.P., and G.C. helped for data collection. X.G. aided in the synthesis of PIM polymers. Q.L. and N.L. supervised and guided this work. H.T., Q.L., and N.L. analyzed all experimental data and wrote the paper. K.G., D.A., and X.Y. helped to write the manuscript.

## Competing interests

The authors declare no competing interests. Received: (will be filled in by the editorial staff). Revised: (will be filled in by the editorial staff). Published online: (will be filled in by the editorial staff)
