## [Peer review file · Nature Communications]

REVIEWER COMMENTS

Reviewer #1 (Remarks to the Author):

The manuscript "High Pt Utilization Enabled by Hydrogen-bonding Microporous Polymer Binders for High-performance Fuel Cells" by Tang et al. describes the design and synthesis of novel catalyst binder materials based on polymers of intrinsic microporosity (PIMs) with strong hydrogen-bonding functionalities for high temperature proton exchange membrane fuel cell. The Pt utilization has been improved greatly and thus the expensive catalyst loading being lowered down by using tetrazole functionalized PIM binder (PIM-Tz). This is an interesting work, even though the catalyst loading of 0.15 mgPt/cm² in the catalyst layer is still a little higher than the target of DOE (0.125 mgPt/cm² for PEMFC). In the light of the achieved performances, I strongly recommend this paper could be published in Nature Communications with minor revisions:

1. How about the cell performance if the catalyst loading is further reduced?
2. Why do you choose tetrazole functionalized PIM with a degree of 75% as the binder material in the catalyst layer? How about fully tetrazole functionalized PIM with tetrazole?

Minor points:

1. Line 36, "lead" should be "leads";
2. Line 57-58, "are difficulty dissolve and diffuse" is wrong;
3. Line 68, "utilization are still necessary" should be "utilization is still necessary";
4. Line 182, There is an excess "s" in "(e.g. higher hydrogen-bonding energy)s";
5. Line 185, "avoding poisoning" is should be "avoiding poisoning";
6. Line 285, "m" should be "μm";
7. Line 326, "is" should be deleted;
8. Line 396, "for" should be deleted;
9. Line 410 "were" in should be changed as "was";
10. Line 419, "used" should be "using";
11. The abbreviation of "mPBI" should be consistent.

Reviewer #2 (Remarks to the Author):

The current manuscript is an interesting one and deals with one of the critical problem for any PEM fuel cell, particularly high-temp PEM based FC. Author has demonstrated the work very well. I recommend publication with the following corrections, which may be included:

1. The author has described Pt loading in some cases as “mgPt cm⁻²” and in some other cases as “mg cm⁻²”. These different representations of Pt loading should be avoided.
2. In line 43, the author has mentioned the catalyst was supported on carbon black but no mention of Pt content in the carbon black was stated. The amount of Pt nanoparticle present in the carbon black plays an important role in the application. The significance can be mentioned in the manuscript and the benefits of choosing Pt content of such %.
3. In line 75, the author has mentioned ‘The best binder’ which may confuse the reader. Please mention it correctly by reconstructing the sentence.
4. In line 88, the author has mentioned ‘ENREF 24’ in the sentence which is not clear. Please recheck it.
5. In line 115, degree of functionalization was mentioned as 100% but no supporting data or the calculation procedure is available. Please mention the instrumentation technique used to determine such factor.
6. In line 117, the molecular weight was observed to decrease 40-53kDa upon post modification of PIM1. Does it represent the degradation of the PIM1?
7. The thermal stability can be checked in terms of thermos gravimetric analysis (TGA) of the samples as well as PA doped samples and can be highlighted.
8. In line 144, the author has mentioned Grotthuss mechanism is taking place. Vehicular mechanism is also possible in some particular cases. Did the author check activation energy at different temperature from the proton conductivity values in order to support the argument?
9. pKa value plays a crucial role in PA doping as well as PA leaching. It would be better if the pKa calculations can be explained in detail from RGC.
10. In line 203, Figure 2 a, pore size distribution curve shows the pore size is above 100 nm (1 nm) but the same is described in line 200 as 0.4 to 20nm. Please clarify the anomaly. The pore size should be below 1nm in order to obtain ultramicroporosity. Please explain.
11. In line 242, Figure 3 b, the units of the four-axis, mentioned in the graph is not clear. Please indicate the assigned values of all the four axes.
12. In line 279, Figure 4 d, the units of the three-axis, mentioned in the graph is not clear. Please indicate the assigned values of all the three axes.

Reviewer #3 (Remarks to the Author):

This communication represents an interesting addition to HT-PEM FC research field. The topic is novel and relevant. In particular, the optimisation of binders for this FC type is often omitted for the favour of membranes. Study covers preparation of binders, their implementation into electrodes followed by detailed characterisation and single cell testing. Authors demonstrated promising results and supported them with sufficient data. However, there are certain issues to be addressed before publication.

General remarks:

There is no heading indicating the start of “Results and Discussion” section.

There is a high number of typing errors in manuscript.

Readability of text is not ideal, especially due to vast number of experiments with various parameters. It is not entirely clear what were conditions and materials used for each experiment shown in figures. Description of figures is often misleading. Authors may consider adding a table, specifying performed single cell experiments. In general, structure of experimental part can be significantly improved and separated into parts corresponding to the ones in results and discussion section.

Authors often discuss in text Pt utilisation, which they determine based on size of Pt nanoparticles in catalyst. However, there are certain problems connected to their experimental procedure. Voltammograms measured with anode as hydrogen pseudo-reference electrode will never provide exact information about electrochemically active surface area on the cathode and absolute values are almost never correct, especially in system with strongly adsorbing electrolyte. Relative comparison is correct, but then the exact info about the catalyst is needed, which is missing in study.

Electrochemical stability of PBI binder is often an issue (10.1149/2.0741506jes). Did authors consider performing electrochemical experiments to verify the stability of PIM-based binders?

Specific comments:

Line 32 – What authors mean by strong adsorption of phosphoric acid molecules at low potentials and acid anions at high potentials? This is essentially wrong, because at low potentials corresponding to HUPD, phosphate anions are mostly removed/reduced to phosphorous acid and at high potentials surface is mostly covered by adsorbed O. This part of text needs clarification according to potential ranges. Authors may consider adding references as for example 10.1021/jp311924q, 10.1016/j.electacta.2015.01.097, 10.1002/celc.201300134.

Line 64 – Have authors considered also binders based on ion-pairs?

Line 227 – Is hydrogen desorption region really well suited for determination of electrochemically active surface area? What about adsorption region?

Line 280, Figure 4 – Results in this figure does not seemingly match the impedance spectra provided in supplementary materials, Figure 6, although it seems that conditions are the same. In Figure 4, fuel cell using PTFE and PBI binders exhibit the lowest HFR, but that is not the case in Supplementary Figure 6, where PTFE has approximately the same resistance as PIM-1 binder. In addition, impedance spectra for PBI and best binder PIM-Tz are almost identical. Ascribing polarisation resistance to mass-transfer related phenomena is essentially incorrect and spectra are not fitted. This presents a major problem, as in present form, these data does not support conclusions. Authors may consider checking their data and fit the spectra. That way, the impact of binder on anodic and cathodic reaction can be evaluated as well.

Line 365 – Was commercial GDE used in work?

Line 413 – it is not clear for which purpose these membranes were prepared. Was it for the determination of permeability or for MEA preparation?

Line 517 – Is there a reason why two different potentiostats were used in study?

Response to Reviewers' comments:

Reviewer #1 (R1):

The manuscript "High Pt Utilization Enabled by Hydrogen-bonding Microporous Polymer Binders for High-performance Fuel Cells" by Tang et al. describes the design and synthesis of novel catalyst binder materials based on polymers of intrinsic microporosity (PIMs) with strong hydrogen-bonding functionalities for high temperature proton exchange membrane fuel cell. The Pt utilization has been improved greatly and thus the expensive catalyst loading being lowered down by using tetrazole functionalized PIM binder (PIM-Tz). This is an interesting work, even though the catalyst loading of 0.15 mgPt/cm² in the catalyst layer is still a little higher than the target of DOE (0.125 mgPt/cm² for PEMFC). In the light of the achieved performances, I strongly recommend this paper could be published in Nature Communications with minor revisions:

A1-0 Response: We thank the reviewer for the positive comments.

1. *How about the cell performance if the catalyst loading is further reduced?*

A1-1 Response: Further reduction of the Pt loading is also studied, which gives much lower cell performance. Enhancement of the FC performance is possible but needs further optimization of the catalyst layer, which will be the future work.

2. *Why do you choose tetrazole functionalized PIM with a degree of 75% as the binder material in the catalyst layer? How about fully tetrazole functionalized PIM with tetrazole?*

A1-2 Response: According to the previous work (Nature Material, 2011, 10, 372-375, DOI:10.1038/NMAT2989), the fully (100%) terazole functionalize PIM has been synthesized which has S_{BET} of 30 m²/g. This material is very brittle due to the abundant intramolecular hydrogen bonds. We have tried to use it as the binder in the catalyst layer for HT-PEMFC, however, its low porosity and hence low gas permeability lead to a poor cell performance. It is therefore that a relatively low (75%) functionalization degree is used in the present work. This comments is addressed by adding te following lines in the revised mansucript:

Line 126-129: "PIM-Tz with DF100% with a BET area of 30 m²/g was initially tested which showed poor performance due to the strong hydrogen bonding and the resultant low porosity or gas permeability. In the following work a relatively low (75%) DF is used for PIM-Tz."

A1-3 Response to Minor points: The manuscript has been polished and small changes throughout the manuscript are made including editorial format and linguistic corrections.

Reviewer #2 (R2):

The current manuscript is an interesting one and deals with one of the critical problem for any PEM fuel cell, particularly high-temp PEM based FC. Author has demonstrated the work very well. I recommend publication with the following corrections, which may be included:

A2-0 Response: We are very grateful to the reviewer for the positive comments.

1. *The author has described Pt loading in some cases as “mgPt cm⁻²” and in some other cases as “mg cm⁻²”. These different representations of Pt loading should be avoided.*

A2-1 Response: The Pt loading has been uniformly expressed as “mg_{Pt} cm⁻²”.

2. *In line 43, the author has mentioned the catalyst was supported on carbon black but no mention of. The amount of Pt nanoparticle present in the carbon black plays an important role in the application. The significance can be mentioned in the manuscript and the benefits of choosing Pt content of such %.*

A2-2 Response: Practical catalysts are with 40-60 wt%Pt loading on carbon. High metal loadings are often used in order to reduce the overall catalyst layer thickness and hence the mass transportation limitation. This information is added to the revised manuscript as follow:

Line 45-46, “The Pt loading is normally in a high metal 40-60 wt%Pt range in order to reduce the over catalyst layer thickness and hence the mass transportation limitation.”

3. *In line 75, the author has mentioned ‘The best binder’ which may confuse the reader. Please mention it correctly by reconstructing the sentence.*

A2-3 Response: The wording “the best binder” has been changed to “the candidate binder” in the revised manuscript.

4. *In line 88, the author has mentioned ‘ENREF 24’ in the sentence which is not clear. Please recheck it.*

A2-4 Response: This is an error and “ ENREF 24” has been deleted.

5. In line 115, degree of functionalization was mentioned as 100% but no supporting data or the calculation procedure is available. Please mention the instrumentation technique used to determine such factor.

A2-5 Response: For PIM-COOH, PIM-Ao and PIM-CONH₂, the degree of the functionalization (DF) is determined by the integral ratio of functional hydrogen to aromatic hydrogen in the polymers via the analysis of ¹H NMR spectra as shown in Supplementary Figure 2. While for PIM-Tz, due to the very strong intermolecular hydrogen bonds in the polymer matrix, it is difficult to obtain the DF by ¹H NMR analysis. Thus, the DF is assessed by the functional conditions e.g. the functional time according to the reported literature. The corresponding discussion has been added in the revised manuscript and ESI as follows:

Line 122-126: “The degree of the functionalization for PIM-CONH₂, PIM-AO and PIM-COOH is determined by the integral ratio of functional hydrogen to aromatic hydrogen in the polymers via the analysis of ¹H NMR spectra as shown in Supplementary Figure 2. For PIM-Tz, the DF is assessed by the reaction conditions according to the literature due to the difficult analysis of ¹H NMR spectra resulting from the strong intermolecular hydrogen bonds in the polymer matrix.²³.....”

Supplementary Figure 2. ¹H NMR spectra of the PIMs.

6. In line 117, the molecular weight was observed to decrease 40-53kDa upon post modification of PIM-1.

Does it represent the degradation of the PIM-1?

A2-6 Response: During the modification process, PIM-1 degrades due to the severe reaction conditions e.g. strong acidity or high temperature as shown in the experimental part in the manuscript, thus leading to a drop of the molecular weight of the functional PIMs. This information is added to the revised manuscript:

Line 129-131: “During the modification process, PIM-1 degrades due to the severe reaction conditions e.g. strong acidity or high temperature as indicated by the slightly decreased molecular weight ranging from 40 to 53 kDa (Table 1).”

7. *The thermal stability can be checked in terms of thermogravimetric analysis (TGA) of the samples as well as PA doped samples and can be highlighted.*

A2-7 Response: We thanked the reviewer for pointing out this valuable point. The thermal stability of the PIMs has been tested and added to Supplementary Figure 3. However the TGA of PA doped samples was not measured due to the possible contamination of the PA to the instrument, The corresponding discussion has been added in the revised manuscript and SI as follows:

Line 131: “..... The thermogravimetric analysis (Supplementary Figure 3) shows that the PIMs are thermally stable enough to withstand high temperatures for HT-PEMFC application.”

Supplementary Figure 3. Thermogravimetric analysis of PIMs.

8. *In line 144, the author has mentioned Grotthuss mechanism is taking place. Vehicular mechanism is also*

possible in some particular cases. Did the author check activation energy at different temperature from the proton conductivity values in order to support the argument?

A2-8 Response: We agree with the reviewer that both Grotthuss and vehicule mechanisms of proton conductivity are present in PA as well as in PA doped PBI membranes. This information is added to the revised manuscript.

Line 158-159: “..... the dominating Grotthuss mechanism of the proton conductivity for 100% PA. In case of PA doped PBI membranes, more than 96% of the Grotthussian conductivity was recently reported.³²”

Similar mechanisms are believed to prevail in the case of PA doped functional PIMs, though the activation energy of the proton conductivity is unavailable in the present work.

9. *pKa value plays a crucial role in PA doping as well as PA leaching. It would be better if the pKa calculations can be explained in detail from RGC.*

A2-9 Response: The pKa calculations is explained in detail from RGC in the experiment part, page 22, line 523-526.

10. *In line 203, Figure 2 a, pore size distribution curve shows the pore size is above 100 nm (1 nm) but the same is described in line 200 as 0.4 to 20nm. Please clarify the anomaly. The pore size should be below 1nm in order to obtain ultramicroporosity. Please explain.*

A2-10 Response: The pore size should be 1 nm to 20 nm. The corresponding discussion has been revised in the manuscript.

11. *In line 242, Figure 3 b, the units of the four-axis, mentioned in the graph is not clear. Please indicate the assigned values of all the four axes.*

A2-12 Response: The assigned values of all the four axis have been added in Figure 3b in the revised manuscript.

12. In line 279, Figure 4 d, the units of the three-axis, mentioned in the graph is not clear. Please indicate the assigned values of all the three axes.

A2-12 Response: The assigned values of all the three axis have been added in Figure 4d in the revised manuscript as below.

Reviewer #3 (R3):

This communication represents an interesting addition to HT-PEM FC research field. The topic is novel and relevant. In particular, the optimisation of binders for this FC type is often omitted for the favour of membranes. Study covers preparation of binders, their implementation into electrodes followed by detailed

characterisation and single cell testing. Authors demonstrated promising results and supported them with sufficient data. However, there are certain issues to be addressed before publication.

A3-0 Response: We appreciated very much the reviewer's positive comments.

General remarks:

1. There is no heading indicating the start of "Results and Discussion" section.

A3-1 Response: The head of "Results and Discussion" has been added in the second part in the manuscript.

2. There is a high number of typing errors in manuscript.

A3-2 Response: See Response to R1-3 above (The manuscript has been polished and small changes throughout the manuscript are made including editorial format and linguistic corrections.)

3. Readability of text is not ideal, especially due to vast number of experiments with various parameters. It is not entirely clear what were conditions and materials used for each experiment shown in figures. Description of figures is often misleading. Authors may consider adding a table, specifying performed single cell experiments. In general, structure of experimental part can be significantly improved and separated into parts corresponding to the ones in results and discussion section.

A3-3 Response: We appreciated the reviewer for this suggestion. A table for specifying the test conditions of the single cell has been added in the ESI and is shown as below, the corresponding discription of the cell test conditions in the Figures has been deleted and modified in the revised manuscript. Moreover, the structure of the experimental part has been separated into parts corresponding to the ones in results and discussion section in the revised manuscript.

Supplementary Table 1, The test conditions of the single cells at 160 °C with no extra humidification and no back pressure.

The cell test conditions using different binder materials in the catalyst layer				
Binder	Membrane	Catalyst loading (mg_{Pt} cm⁻²)^a	Binder content (wt%)^b	Flow rate of H₂-O₂ (sccm)
PIM-Tz	mPBI/PA (78 μm)	0.500	20	30c
PIM-COOH	mPBI/PA (80 μm)	0.501	20	30

PIM-AO	m PBI/PA (79 μ m)	0.505	20	30
PIM-CONH₂	m PBI/PA (81 μ m)	0.500	20	30
PIM-1	m PBI/PA (81 μ m)	0.488	20	30
PTFE	m PBI/PA (80 μ m)	0.506	20	30
mPBI	m PBI/PA (77 μ m)	0.510	20	30

The cell test conditions using PIM-Tz binder in the catalyst layer

PIM-Tz	m PBI/PA (78 μ m)	0.501	5	30
PIM-Tz	m PBI/PA (79 μ m)	0.499	10	30
PIM-Tz	m PBI/PA (80 μ m)	0.500	20	30
PIM-Tz	m PBI/PA (78 μ m)	0.700	10	30
PIM-Tz	m PBI/PA (77 μ m)	0.350	10	30
PIM-Tz	m PBI/PA (78 μ m)	0.150	10	30

The cell test conditions using PIM-Tz binder in the catalyst layer with H₂-air fuel

PIM-Tz	m PBI/PA (78 μ m)	0.350	10	30/100
PIM-Tz	m PBI/PA (77 μ m)	0.150	10	30/100

^a Catalyst loading in both cathode and anode. ^b Binder content refers to the binder as a percentage of the solid in the catalyst slurry.

4. Authors often discuss in text Pt utilization, which they determine based on size of Pt nanoparticles in catalyst. However, there are certain problems connected to their experimental procedure. Voltammograms measured with anode as hydrogen pseudo-reference electrode will never provide exact information about electrochemically active surface area on the cathode and absolute values are almost never correct, especially in system with strongly adsorbing electrolyte. Relative comparison is correct, but then the exact info about the catalyst is needed, which is missing in study.

A3-5 Response: We do agree with the view that the cyclic voltammograms provide a rough estimation of the electrochemically active surface area (ECSA) of the cathode catalysts. The “Pt utilization“ based on the ECSA is therefore a poorly defined term. We have therefore revised the manuscript by using “the catalyst performance” instead of “the Pt utilization” in a general sense. In specific cases the term “the Pt-mass specific peak power density, which is directly obtainable from the cell performance and electrode Pt loading, is used instead of “the Pt utilization”. However, the content of the electrochemically active surface area (ECSA) is kept for relative/qualitative comparison of electrodes with different binder materials.

Accordingly, the title of the manuscript has been changed as “Low Pt Loading for High-performance Fuel Cell Electrodes Enabled by Hydrogen-bonding Microporous Polymer Binders”.

In addition, more specific information on the catalyst is given in the Experimental Materials part, Line 384:

“.....Pt/C (HPT040, 38.90-41.10 wt% Pt, particle size < 4.5 nm).....”

5. *Electrochemical stability of PBI binder is often an issue (10.1149/2.0741506jes). Did authors consider performing electrochemical experiments to verify the stability of PIM-based binders?*

A3-5 Response: We thanked the reviewer for the valuable suggestion, which is very helpful for our future work. The work by Kondratenko et al. [10.1149/2.0741506jes] is added as reference 42, as it has a good value in addressing an important binder issue – the Pt stability (dissolution). Concerning the stability of binder materials themselves, other two references are added (10.1016/j.jpowsour.2021.230642 and 10.1016/j.jpowsour.2020.228521), which have shown that the ether linkage in the binder backbone may suffer scission while the tetrazole group should be very stable under the fuel cell thermal/acidic conditions. It seems conclusive that the PIM-Tz binder is electrochemically stable. The corresponding comments and literatures are addressed by adding the following line in the revised manuscript:

Line 293: “...**Though the presence of PBI as the catalyst binder is reported to influence the Pt dissolution during the fuel cell operation,⁴² the electrochemical stability of the tetrazole groups of the PIM binders does not seem an issue.^{43,44}**”

42 **Kondratenko, M. S. et al. Degradation of high temperature polymer electrolyte fuel cell cathode material as effected by polybenzimidazole. 162 (6), F587-F595, 10.1149/2.0741506jes (2015).**

43 **Tang, H. et al. Properties and stability of quaternary ammonium-biphosphate ion-pair poly(sulfone)s high temperature proton exchange membranes for H₂/O₂ fuel cells. *J. Power Sources*, **475**, 228521, doi:10.1016/j.jpowsour.2020.228521 (2020).**

44 **Tang, H. et al. On the stability of imidazolium and benzimidazolium salts in phosphoric acid based fuel cell electrolytes. *J. Power Sources*, **515**, 230642, doi:10.1016/j.jpowsour.2020.228521 (2022).**

Specific comments:

6. *Line 32 – What authors mean by strong adsorption of phosphoric acid molecules at low potentials and acid anions at high potentials? This is essentially wrong, because at low potentials corresponding to HUPD, phosphate anions are mostly removed/reduced to phosphorous acid and at high potentials surface is mostly covered by adsorbed O. This part of text needs clarification according to potential ranges. Authors may consider adding references as for example 10.1021/jp311924q, 10.1016/j.electacta.2015.01.097, 10.1002/celc.201300134.*

A3-6 Response: We appreciate the comment and have corrected the error. In the revised manuscript two new references are added and the following text is added:

Line 32-34: “**In the presence of phosphoric acid electrolyte, the Pt surface involves the strong adsorption**”

of acid molecules (H_3PO_4) in the low potential range (300-400 mV) and acid anions (H_2PO_4^-) at intermediate (700-800 mV) potential range during the cell operation at 160 °C.⁵⁻⁸.....”

- 6 Kaserer, S. *et al.* Analyzing the influence of H_3PO_4 as catalyst poison in high temperature PEM fuel cells using in-operando X-ray absorption spectroscopy. *J. Phys. Chem. C*, **117**, 6210-6217, doi:10.1021/jp311924q (2013).
- 7 Doh, W. H. *et al.* Electrochemistry of phosphorous and hypophosphorous acid on a Pt electrode. *Electrochimica Acta*, **160**, 214–218, 10.1016/j.electacta.2015.01.097 (2015).
- 8 Peron, J. *et al.* Scanning photoelectron microscopy study of the Pt/phosphoric-acid-imbibed membrane interface under polarization. *ChemElectroChem*, **1**, 180 – 186, doi: 10.1002/celec.201300134 (2014).

7. Line 64 – Have authors considered also binders based on ion-pairs?

A3-7 Response: According to our previous work (10.1016/j.jpowsour.2021.230642 and 10.1016/j.jpowsour.2020.228521), the (benz)imidazoliums are more stable than the quaternary ammoniums in simulated operation conditions of HT-PEMFC, thus the binders based on (benz)imidazolium biphosphate ion-pairs are under the study in our lab and will be published in the near future. This information is however not included in the present manuscript.

8. Line 227 – Is hydrogen desorption region really well suited for determination of electrochemically active surface area? What about adsorption region?

A3-8 Response: For determination/estimation of ECSA, the hydrogen desorption region is more reliable. The reversal potential is a critical issue for the hydrogen adsorption region as the hydrogen evolution reaction may be involved at potentials near the reversible hydrogen reduction potential (0.0 V). In case of phosphoric acid electrolyte or Pt alloy catalysts, the CO stripping method is more appropriate, which is however not available for the present study. No action is taken in connection to this comment.

9. Line 280, Figure 4 – Results in this figure does not seemingly match the impedance spectra provided in supplementary materials, Figure 6, although it seems that conditions are the same. In Figure 4, fuel cell using PTFE and PBI binders exhibit the lowest HFR, but that is not the case in Supplementary Figure 6, where PTFE has approximately the same resistance as PIM-1 binder. In addition, impedance spectra for PBI and best binder PIM-Tz are almost identical. Ascribing polarisation resistance to mass-transfer related phenomena is essentially incorrect and spectra are not fitted. This presents a major problem, as in present form, these data does not support conclusions. Authors may consider checking their data and fit the spectra. That way, the impact of binder on anodic and cathodic reaction can be evaluated as well.

A3-9 Response: In a qualitative manner we understood the EIS in the following ways: 1) The HFR

originates from the ohmic resistance of the membrane, which is determined by the amount of the doping acid. When a fuel cell is assembled and activated (during the breakin period) the doping acid transfers from the membrane to the catalyst layer. For hydrophobic (with weaker acid affinity) binders (PTFE and PIM-1) less acid is absorbed by the catalyst layer i.e. more acid remains in the membrane. As a result, the HFR due to the membrane resistance will be smaller while the kinetic resistance will be larger. The same arguments apply to the hydrophilic binders (PIM-Tz, mPBI, etc) with opposite behaviors. This is consistent with both the I-V curves (Figure 4) and EIS (Supplementary Figure 8 in the revised SI).

This is further clarified in the revised manuscript: [lines 269]: „... The HFR originates from the ohmic resistance of the membrane, which is determined by the amount of the doping acid. When a fuel cell is assembled and activated the doping acid transfers from the membrane to the catalyst layer.⁴⁰ ...” As a consequence such MEAs exhibit a relatively larger membrane resistance i.e. HFR. ...

10. Line 365 – Was commercial GDE used in work?

A3-10 Response: The commercial GDEs from Danish Power Systems (now Blue World Technologies) were used in this work, and the H₂-air cell performance was listed in the Supplementary Table 3 in the SI.

11. Line 413 – it is not clear for which purpose these membranes were prepared. Was it for the determination of permeability or for MEA preparation?

A3-11 Response: Yes, the purpose to prepare PIMs membranes is for determination of the gas permeability of the bulk polymer materials themselves, which is indicative to the gas permeability of the corresponding GDEs.

12. Line 517 – Is there a reason why two different potentiostats were used in study?

A3-12 Response: All the electrochemical performances were measured by a Bio-Logic VSP-300 potentiostat. The fuel cell workstation (Smart 2-WonATech Inc., Korea) was employed to check the repeatability of the cell performance. The corresponding description has been changed in the revised manuscript as follows:

Line 597: “.....A fuel cell workstation (Smart 2-WonATech Inc., Korea) was employed to check the repeatability of the cell performance.....”

REVIEWERS' COMMENTS

Reviewer #1 (Remarks to the Author):

The revision article has been revised in detail in accordance with the revision opinions, which can be accepted and published at present.

Reviewer #2 (Remarks to the Author):

Author has revised the manuscript quite well by responding to all the questions raised by the reviewers

Reviewer #3 (Remarks to the Author):

I would like to thank authors for addressing all comments, their effort is well-appreciated. In my opinion, reviewed version of manuscript meets high standards of journal and should be published.